# SHORT REPORT

# TBC1D15 functions as an Arl4D GAP and promotes the mitochondrial translocation of Arl4D for organelle homeostasis

Chia-Tang Chen[1,2,*], Tsai-Jung Liu[1,2,*], Shin-Jin Lin[1,2,*], Ting-Wei Chang[1,2] and Fang-Jen S. Lee[1,2,3,‡]

## ABSTRACT

ADP-ribosylation factor-like 4D (Arl4D), a Ras small GTPases superfamily member, plays crucial roles in membrane trafficking, cytoskeletal remodeling and cell migration. GDP-bound Arl4D has previously been shown to locate at the mitochondria and alter mitochondrial morphology and activity; however, how the nucleotide-binding state and mitochondrial targeting of Arl4D is regulated had remained unclear. We now discover that TBC1D15, a well-known Rab7 GTPase-activating protein (GAP), functions also as an Arl4D GAP to promote Arl4D mitochondrial targeting. We initially show that GDP-bound Arl4D translocates to the mitochondria under serum starvation and affects mitochondrial homeostasis. We also show that TBC1D15 interacts with Arl4D through the TBC domain and promotes GTP hydrolysis of Arl4D. Knockdown of TBC1D15 leads to an increase in Arl4D activity and decreased Arl4D mitochondrial translocation under serum starvation. These findings support the hypothesis that TBC1D15 acts as an Arl4D GAP and reveal a new role for this GAP in modulating mitochondrial homeostasis.

KEY WORDS: ADP-ribosylation factor, Arl4, GTPase, GTPase activating protein, Mitochondria

## INTRODUCTION

ADP-ribosylation factor (Arf) and Arf-like (Arl) proteins, which belong to the Ras superfamily of small GTPases, play a central role in regulating various cellular processes, such as membrane trafficking, organelle integrity and cytoskeletal remodeling. Similar to other small GTPases, Arf and Arl proteins are modulated by guanine nucleotide exchange factors (GEFs) and GTPase-activating proteins (GAPs), which regulate their activity by promoting the release of bound GDP or by catalyzing GTP hydrolysis (Qu et al., 2019; Takai et al., 1992; Wennerberg et al., 2005). Recent studies have suggested that GTP-bound Arl4D facilitates the activation of Arf6 by recruiting ARNO (also known as CYTH2) to the plasma membrane and regulates microtubule growth by forming complexes with EB1 (also known as MAPRE1) (Li et al., 2007; Lin et al., 2020). However, the GEFs or GAPs for Arl4D, which would be crucial for investigating

[1]Institute of Molecular Medicine, College of Medicine, National Taiwan University, 10002 Taipei, Taiwan. [2]Department of Medical Research, National Taiwan University Hospital, 10002 Taipei, Taiwan. [3]Center of Precision Medicine, College of Medicine, National Taiwan University, Taipei 10002, Taiwan.
*These authors contributed equally to this work

‡Author for correspondence (fangjen@ntu.edu.tw)

F.-J.S.L., 0000-0002-2167-2426

its physiological functions and mechanical regulation, have not yet been identified.

Mitochondria play a central role in various cellular processes, such as lipid metabolism, apoptosis and energy production (Spinelli and Haigis, 2018; Tábara et al., 2025). Although most wild-type Arl4D is active and resides on the plasma membrane, our previous study has shown that the nucleotide-binding-defective Arl4D mutant (Arl4D-T35N) localizes to the mitochondria and causes mitochondrial depolarization and fragmentation (Jacobs et al., 1999; Li et al., 2007, 2012). Mitochondria membrane depolarization triggers the degradation of mitochondria through mitophagy (Onishi et al., 2021), whereas fragmentation indicates increased fission events, which is crucial for removing damaged mitochondria and maintaining mitochondrial quality and content, contributing to mitochondrial homeostasis (Eisner et al., 2018). This hints that inactive Arl4D plays a role in regulating mitochondrial homeostasis.

In this study, we identify TBC1D15 as an Arl4D GAP that regulates its mitochondria targeting. We observed that TBC1D15 increases the intrinsic GTP hydrolysis of Arl4D and increases the mitochondrial localization of Arl4D. TBC1D15 depletion significantly reduced mitochondrial Arl4D level under serum starvation. Further study shows that depleted Arl4D compromised hypoxia-induced mitophagy under serum starvation. Our findings not only identify TBC1D15 acts as a novel GAP for Arl4D but also reveal a new role for this GAP in mitochondrial homeostasis.

## RESULTS AND DISCUSSION

### The GDP-bound Arl4D is increased under serum starvation and translocates to mitochondria

We found that Arl4D translocates to the mitochondria as cell culture duration increases up to 96 h (Fig. 1A). Over this period, cell density and stress accumulate while glucose and serum growth factors levels in the medium decline. To determine the factors driving the mitochondrial translocation of Arl4D, we tested three conditions – serum starvation, increased cell density and glucose deprivation. Our findings show that only serum starvation significantly enhanced the translocation of Arl4D into mitochondria (Fig. 1B). Given that we previously identified that GDP-bound Arl4D localizes to the mitochondria, we further investigated the impact of serum starvation on the GTP-binding status of Arl4D. We then measured Arl4D activity using a Pak–PBD pulldown assay, which specifically isolates the active GTP-bound form of Arl4 proteins (Chen et al., 2020). This experiment showed that serum starvation significantly decreased the amount of active Arl4D in HeLa cells (Fig. 1C). To investigate whether translocation of Arl4D to mitochondria is induced by GDP binding under serum deprivation, we overexpressed wild-type Arl4D (Arl4D-WT), the constitutively active form (Q80L), and inactive form (T35N) in HeLa cells under normal and serum-deprived conditions. We found that Arl4D-T35N localized to mitochondria regardless of culture conditions, whereas Arl4D-WT translocated to mitochondria only under serum starvation conditions. Conversely,

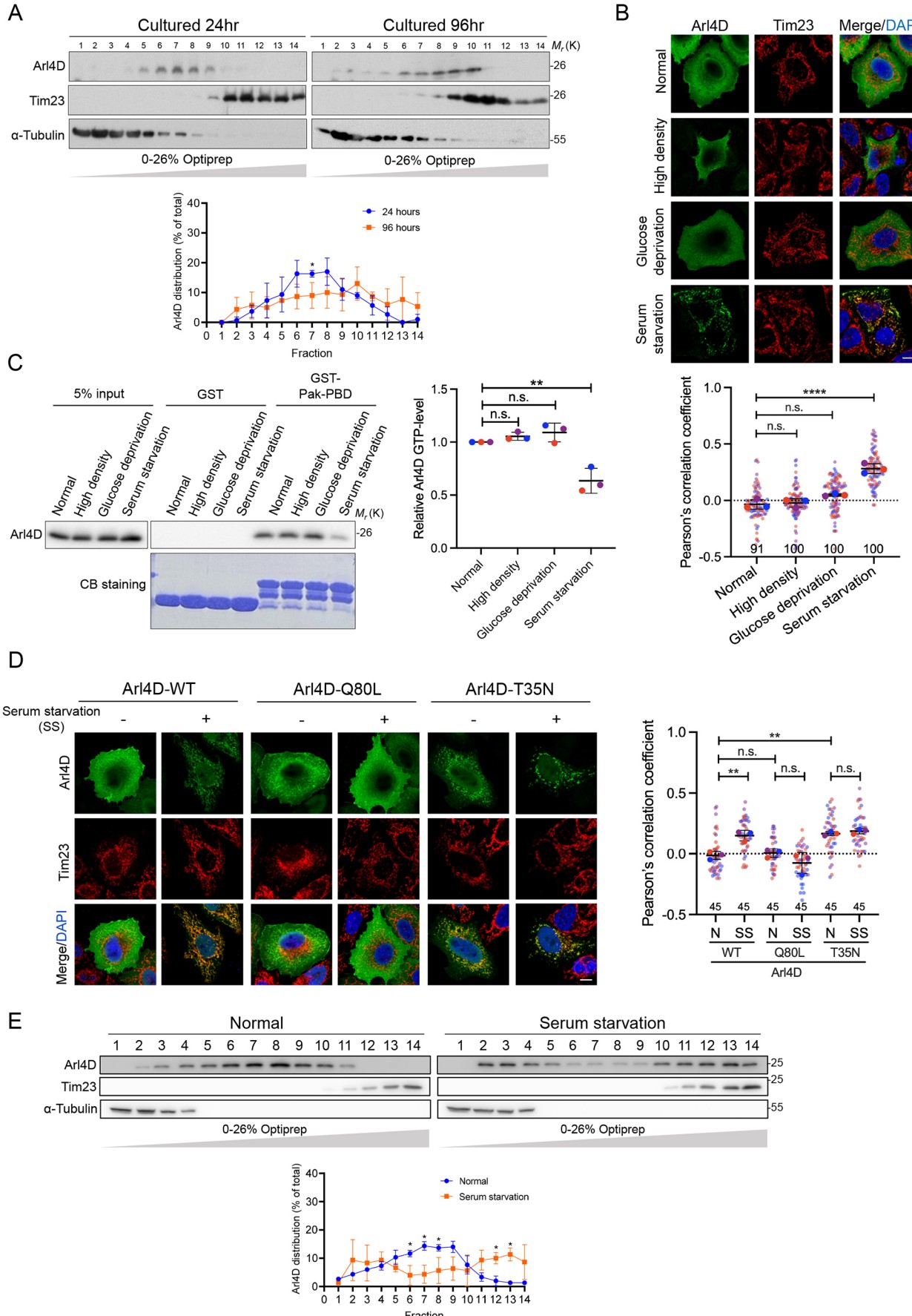

**Fig. 1.** See next page for legend.

**Fig. 1. Serum starvation induces inactivation and mitochondria targeting of Arl4D.** (A) Lysates from HeLa cells cultured for indicated duration were fractionated by gradient centrifugation and analyzed by immunoblotting. Fractions with signals for Tim23 and α-tubulin indicate the mitochondria and cytosol fraction, respectively. (B) Arl4D-overexpressing HeLa cells were cultured in either high-cell density conditions, or glucose-deprivation or serum starvation medium for 24 h before fixing and staining with anti-Arl4D (green) and anti-Tim23 (red) antibodies, and DAPI (blue; nucleus). Quantification of colocalization of Arl4D and Tim23 is represented by the Pearson's correlation coefficient expressed as mean±s.d. (*n*=3; 90 cells were analyzed per group) and analyzed by one-way ANOVA with Tukey's post hoc multiple comparison test to determine *P*-value. (C) HeLa cells conditioned as described in B were lysed and submitted to Pak–PBD pulldown assay. The amount of active Arl4D pulled down by Pak-PBD quantified by densitometric methods and presented as mean±s.d. (*n*=3) and analyzed by one-way ANOVA with Tukey's post hoc multiple comparison test. (D) HeLa cells transfected with Arl4D-WT, -Q80L or -T35N plasmids were cultured for 24 h under normal conditions (N) or in serum starvation (SS) medium before staining with anti-Arl4D (green) and anti-Tim23 (red) antibodies, and DAPI (blue). Colocalization of Arl4D and Tim23 was quantified with Pearson's correlation coefficient. Data are presented as mean±s.d. (*n*=3; 45 cells per group) and one-way ANOVA with Tukey's post hoc multiple comparison test was applied to determine *P*-value. (E) Lysates of HeLa cells cultured in normal or serum-free medium for 24 h were fractionated by gradient centrifugation and analyzed by immunoblotting. For A and E, the percentage of total protein in each fraction is quantified (mean±s.d., *n*=3); unpaired two-tailed t-test was applied to determine *P*-value. *$P<0.05$, **$P<0.01$, ****$P<0.0001$; n.s., not significant. CB, Coomassie Blue. Scale bars: 10 μm.

Arl4D-Q80L barely localized to mitochondria under either condition (Fig. 1D). Additionally, fractionation experiments revealed that endogenous Arl4D translocated to mitochondria during 24 h of serum starvation, evidenced by a shift to heavier fractions that colocalized with Tim23 (also known as TIMM23) (Fig. 1E). According to the above data, we found that the serum starvation signal regulates Arl4D activity and mitochondria localization.

### Reduction of mitochondrial membrane potential leads to relocation of GDP-bound Arl4D to mitochondria

As mitochondrial membrane potential (ΔΨm) decreases under serum starvation (Kajiya et al., 2009; Papucci et al., 2003), we asked whether ΔΨm is key signal to regulate Arl4D mitochondria targeting. We used carbonyl cyanide m-chlorophenylhydrazone (CCCP) to induce a reduction in mitochondrial membrane potential (Ganote and Armstrong, 2003) and found that CCCP reduces the amount of active Arl4D (Fig. S1A) and promotes the translocation of Arl4D to the mitochondria in both HeLa and MDA-MB-231 cell lines, confirming the response is not cell type specific (Fig. S1B,C). Moreover, we found that Arl4A, Arl4C and Arl4D-Q80L did not localize to mitochondria under these conditions (Fig. S1D–F). These results support the idea that the reduction of mitochondrial membrane potential by CCCP treatment increases GTP hydrolysis of Arl4D and induces its mitochondria localization.

### Mitochondria-residing Arl4D facilitates hypoxia-induced mitophagy

Given that mitochondria depolarization induces clearance of dysfunctional mitochondria by mitophagy (Onishi et al., 2021), we speculate that Arl4D might play a role in the elimination of unhealthy mitochondria and thereby influence mitochondrial homeostasis. The mitochondrial fragmentation we previously observed upon Arl4D-T35N overexpression indicates an imbalance in mitochondrial dynamics (Li et al., 2012), which could result from either accelerated fission or inhibited fusion, whereas mitochondria from cells overexpressing Arl4D-WT and Arl4D-Q80L remained tubular

(Fig. S2A). As mitochondria fission and fusion, similar to mitochondrial depolarization, are also important for the balance of healthy mitochondria (Youle and Van Der Bliek, 2012), we investigated whether Arl4D mediates mitophagy to control mitochondrial homeostasis. Although serum starvation alone did not induce significant mitophagy, we observed upregulated mitophagy counts in cells cultured under both hypoxic and serum-starved conditions, and further depletion of Arl4D abolished this upregulation (Fig. S2B). To directly assess whether mitochondria-located Arl4D regulates hypoxia-induced mitophagy, we measured mitophagy counts in cells overexpressing Arl4D-T35N and found that those cells showed increased mitophagy events compared to the control group (Fig. S2C). Together, our results show that mitochondrial translocation of Arl4D mediates hypoxia-induced mitophagy.

### Interactome suggests TBC1D15 as a novel GAP protein for Arl4D

We hypothesized that an unknown GAP protein regulates the activity of Arl4D and facilitates its mitochondrial translocation. A yeast two-hybrid assay was performed, and among the interactome, we identified TBC1D15, a known GAP for Rab7 (herein referring to Rab7a) (Peralta et al., 2010; Zhang et al., 2005) as a candidate (Fig. 2A,B). TBC1D15 amino acids 303–675 was found to interact selectively with Arl4D but not with other Arl proteins (Fig. 2C). The binding specificity was confirmed by co-immunoprecipitation (co-IP) in cultured cells among Arl4 family (Fig. 2D) and the interaction was also verified with endogenous levels of Arl4D (Fig. 2E). Noticeably, the interaction was confirmed to be dependent on Arl4D binding to GTP in the pulldown assays, which is in accordance with the binding preference of a small GTPase with its GAP protein (Fig. 2F). Furthermore, we narrowed down that the GAP catalytic domain of TBC1D15, the TBC domain (326–562), was sufficient to bind Arl4D (Fig. 2G), and that deletion of TBC domain severely compromised their interaction (Fig. 2H). Overall, these results reveal that TBC1D15 could be a novel GAP protein for Arl4D.

### TBC1D15 regulates Arl4D localization to the plasma membrane

TBC1D15 interacting with the active form of Arl4D strengthens the possibility that TBC1D15 serves as a GAP protein for Arl4D. We next examined whether TBC1D15 colocalizes with the active form of Arl4D at the plasma membrane. We co-expressed Arl4D-WT, -Q80L and the myristoylation-deficient mutant -G2A with Myc–TBC1D15 in HeLa cells (Harroun et al., 2005; Li et al., 2012). We found that TBC1D15 only colocalizes with the GTP-locked mutant of Arl4D at the plasma membrane (Fig. S3A), corresponding to the TBC1D15 binding preference of GTP-bound Arl4D. In contrast, recruitment of TBC1D15 to the mitochondria mediated by Fis1 overexpression (Onoue et al., 2013) does not induce mitochondrial targeting of Arl4D, supporting a model where the Arl4D–TBC1D15 interaction occurs at a non-mitochondrial location (Fig. S3B). We further tested whether depletion of this GAP protein enhanced the activity of Arl4D. Data showed that knockdown of TBC1D15 resulted in increased localization of Arl4D to the plasma membrane, similar to where GTP-bound Arl4D localizes (Fig. S3C,D), indicating an increase in GTP-bound Arl4D upon TBC1D15 depletion. These results support that TBC1D15 might act as a GAP protein to regulate Arl4D activity and localization.

### TBC1D15 exerts GAP activity towards Arl4D

We purified recombinant TBC1D15 amino acids 303–675 containing the TBC domain, and examined GTP hydrolysis of

Journal of Cell Science

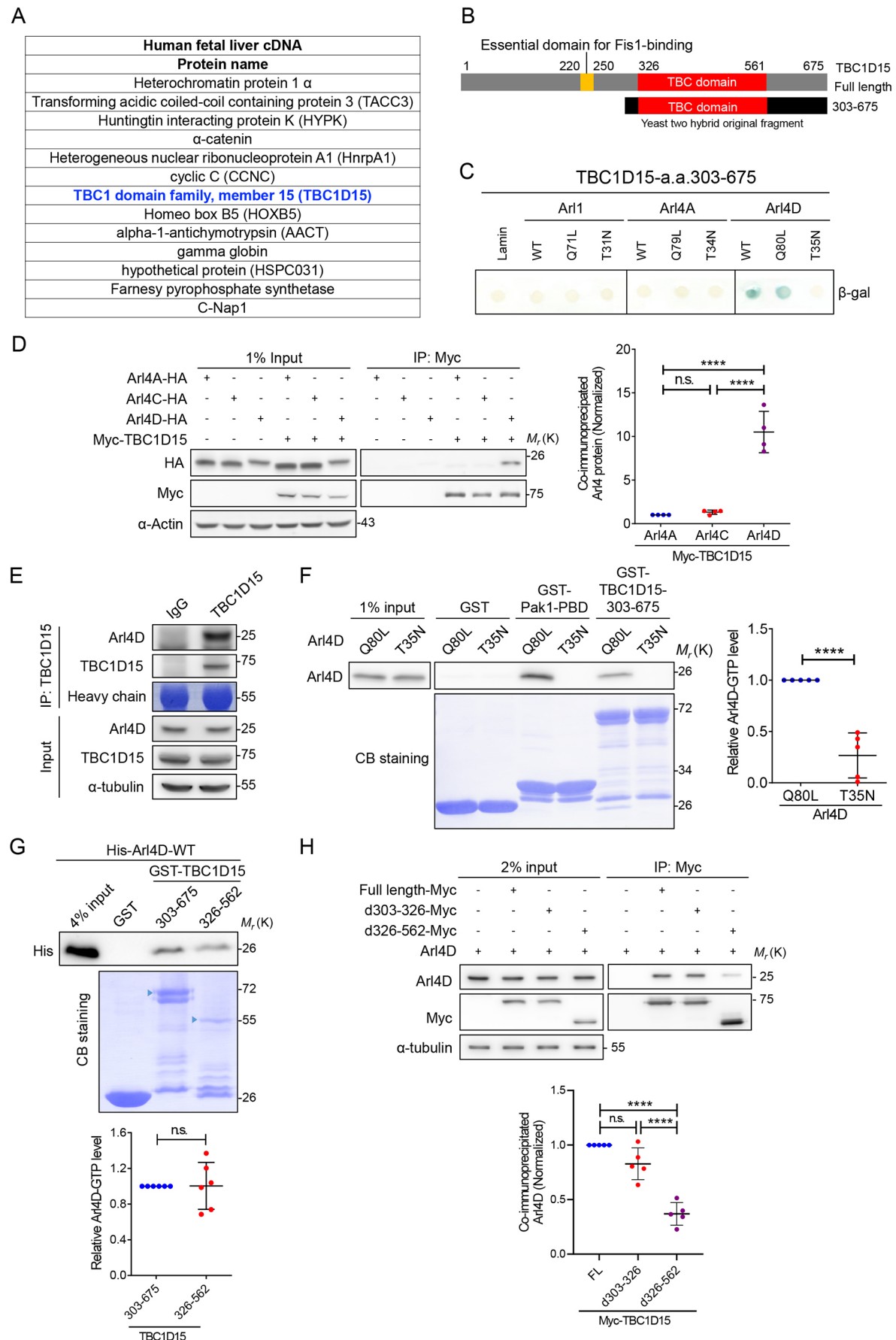

**Fig. 2.** See next page for legend.

**Fig. 2. Arl4D interacts with TBC1D15 via its TBC domain.** (A) List of candidate Arl4D interactors identified by yeast-two hybrid screening of a human fetal liver cDNA library. (B) Schematic representation of the TBC1D15 structure. The schematic shows the region of TBC1D15 (TBC1D15 303–675) that was previously analyzed for interaction with Arl4D in yeast two-hybrid screening. (C) Yeast strain L40 was co-transformed with the TBC1D15 fragment fused to the Gal4 activation domain and the indicated Arl proteins fused to the LexA DNA-binding domain. Expression of β-galactosidase shows a positive interaction. (D) Lysates were prepared from HeLa cells co-expressing Arl4A, Arl4C or Arl4D tagged with HA and Myc–TBC1D15 and co-immunoprecipitated with Myc trap magnetic beads. Bound proteins were analyzed by immunoblotting and quantification was performed by one-way ANOVA with Tukey's post hoc multiple comparison test. Data are presented as mean±s.d. (E) Endogenous TBC1D15 was pulled down with Protein G beads in HeLa cell extracts to assess binding affinity of endogenous Arl4D. IgG serves as negative control. Image shown is representative of one experiment. (F) HeLa cells transfected with Arl4D-Q80L or T35N were lysed and incubated with GST–Pak–PBD or GST–TBC1D15 303-275 in pulldown assays to assess the amount of active Arl4D. Input and pulled down proteins were analyzed by immunoblotting and quantified results were analyzed by unpaired two-tailed Student's t-test. (G) In vitro binding of GST, GST–Pak1–PBD and GST–TBC1D15 303–675 with His–Arl4D. His–Arl4D pulled down by GST-fused proteins were subjected to immunoblotting. Quantified results were analyzed by one-way ANOVA with Tukey's post hoc multiple comparison test. (H) HeLa cells co-transfected with Arl4D–HA and Myc–TBC1D15 fragments were subjected to co-immunoprecipitation and analyzed as described in C. All quantitative results are n=4 to 6; ****P<0.0001; n.s., not significant. CB, Coomassie Blue; FL, full length.

recombinant Arl4D-GTP *in vitro* to provide direct evidence of TBC1D15 acting as a GAP for Arl4D. The EnzChek phosphate assay was used to monitor phosphate release from GTP hydrolysis. Using Rab7, a known substrate of TBC1D15, as a positive control, we found that TBC1D15 enhanced the GTP hydrolysis of Arl4D (Fig. 3A). Furthermore, co-incubation of Arl4D with TBC1D15 D397A, a GAP activity-dead mutant (Rak et al., 2000; Yamano et al., 2014), partially restored GTP hydrolysis of Arl4D compared to TBC1D15 WT (Fig. 3B). These data indicate that TBC1D15 has comparable GAP activity toward Rab7 and Arl4D. We then sought to confirm the *in vivo* regulation of cycling between GTP- and GDP-bound Arl4D by TBC1D15. Using GST–Pak1–PBD to pull down GTP-Arl4D, we found cells expressing TBC1D15 WT but not those expressing D397A mutant had a reduced level of active Arl4D (Fig. 3C), showing that the regulation of GTP-bound Arl4D depends on the GAP activity of TBC1D15 in cultured cells. AS we had identified serum starvation as a crucial signal to turn downregulate Arl4D activity, we further tested whether TBC1D15 is the GAP involved in this condition. We found that depleting TBC1D15 under serum starvation partially restored Arl4D activity as the control groups (Fig. 3D). This result shows that GAP activity of TBC1D15 toward Arl4D is controlled by the stimuli in serum.

## TBC1D15 GAP activity is required for Arl4D mitochondrial translocation
As we discovered TBC1D15 as a GAP for Arl4D, we asked whether TBC1D15 contributes to the Arl4D to mitochondrial localization under serum starvation. Upon knocking down TBC1D15, we found the percentage of cells with Arl4D mitochondrial localization dropped even under normal conditions (Fig. 4A). Even though serum starvation raised the mitochondrial translocation of Arl4D to ~60%, TBC1D15 depletion significantly reduced mitochondrial Arl4D localization to ~40%, and this observation was also confirmed with fractionation assays assessing endogenous Arl4D (Fig. 4B). Given that TBC1D15 is

also the GAP for Rab7, an established mitophagy regulator, we tested whether Arl4D translocation was an indirect effect of Rab7 inactivation. However, siRNA-mediated knockdown of Rab7 had no effect on Arl4D localization, ruling out the involvement of Rab7 in this process (Fig. S4A). Besides, TBC1D15 depletion inhibited the translocation of Arl4D at mitochondria under CCCP treatment, indicating that TBC1D15 triggers Arl4D transporting to depolarized mitochondria (Fig. S4B). We further tested whether the GAP activity is required for Arl4D mitochondria translocation under serum starvation. We expressed siRNA-resistant TBC1D15 WT (TBC1D15^Res-WT) and D397A (TBC1D15^Res-D397A) in TBC1D15-depleted cells, and found that mitochondrial localization of Arl4D could be rescued by TBC1D15^Res-WT but not by TBC1D15^Res-D397A (Fig. 4C). These results revealed that TBC1D15 GAP activity upregulated by serum deprivation induces Arl4D mitochondrial translocation.

Arf GAP proteins exhibit a conserved structural motif known as the 'arginine finger', which plays a crucial role in GTP hydrolysis (Donaldson and Jackson, 2011; East et al., 2012; Sztul et al., 2019). Notably, this structural feature is also present in ELMOD2 and RP2, which function as GAPs for Arl2/3 and Arl3, respectively (Sztul et al., 2019; Veltel et al., 2008). TBC proteins possess an intriguing dual-finger system composed of a conserved 'arginine finger' and a 'glutamine finger', which enhances the efficiency of GTP hydrolysis in small GTPases (Pan et al., 2006). This supports the possibility that TBC1D15 could function as a GAP for structurally related substrates.

Downstream of TBC1D15, Rab7 also regulates mitophagy by mediating the formation of autophagosomes (Tábara et al., 2025). Our results in hypoxia-induced mitophagy suggest that Arl4D participates in the mitophagy process from another independent axis. The mitophagy process mediated by TBC1D15 and Rab7 is downstream of PINK1 and Parkin (Yamano et al., 2014), which are not present in our cell model HeLa cells (Denison et al., 2003); the finding that mitochondrial translocation of Arl4D is unaffected upon Fis1 overexpression or after Rab7 depletion also supports the hypothesis that Arl4D mediates mitophagy in a pathway distinct from those shown previously. Although further studies are needed to elucidate the detailed mechanism of how Arl4D is involved in the mitophagy process, our findings support the idea that Arl4D participates in PINK1-Parkin-independent mitophagy pathways to regulate mitochondrial homeostasis.

In summary, we report that the regulatory mechanism of Arl4D translocation into mitochondria is mediated by TBC1D15 GAP activity. Serum starvation or CCCP treatment serve as environmental cues for TBC1D15 GAP activity, which induces GTP hydrolysis of Arl4D and leads to its detachment from the plasma membrane to mitochondria (Fig. 4D). TBC1D15, a known GAP for Rab7, is now uncovered as the first and novel GAP for Arl4D after 30 years since Arl4 proteins were identified.

## MATERIALS AND METHODS
### Cell culture and transfection
HeLa cells (CCL-2) and MDA-MB-231 cells (HTB-26), purchased from American Type Culture Collection (ATCC, Manassas, VA, USA), were cultured in DMEM (Cytiva, SH30003.03). The medium was supplemented with 10% fetal bovine serum (35-010-CV, Corning), penicillin and streptomycin (Invitrogen) in a humidified incubator with 5% $CO_2$ at 37°C. For MDA-MB-231 cells, DMEM was supplemented with MEM non-essential amino acids solution (Gibco) as suggested by the manufacturer's protocol. Transient transfection was performed using Lipofectamine 2000 reagent according to the manufacturer's protocol (Invitrogen).

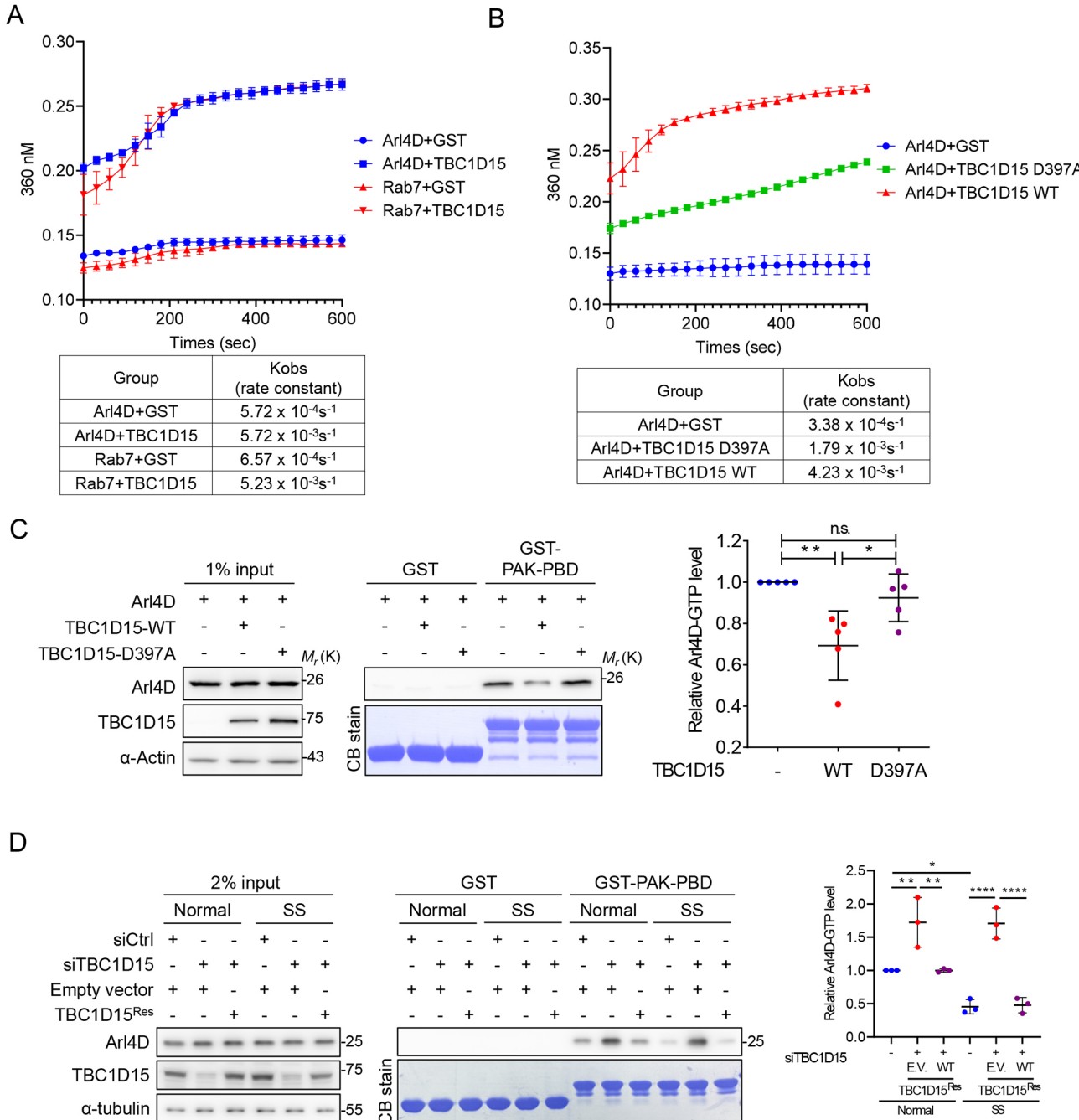

**Fig. 3. TBC1D15 exhibits GAP activity for Arl4D.** (A,B) Measurements of GTP hydrolysis in a GAP assay for Rab7 or Arl4D co-incubated with GST or GST–TBC1D15 protein (A) and Arl4D co-incubated with GST or GST–TBC1D15-WT or -D397A (B). The GAP activity was assessed by EnzChek phosphate assay. $K_{obs}$ (rate constants) are listed in tables below. (C) HeLa cells transfected with the indicated Arl4D–HA and TBC1D15 plasmids were subjected to a Pak–PBD pulldown assay to determine the level of activated Arl4D in HeLa cells. (D) HeLa cells treated with indicated siRNAs and plasmids were cultured in normal or serum-free (SS) DMEM for 16 h. Collected lysate later underwent a Pak–PBD pulldown assay. The input and pulldown proteins in C and D were analyzed by immunoblotting. Quantified data are presented as mean±s.d. and were analyzed by one-way ANOVA with Tukey's post hoc multiple comparison test. All quantitative results are $n$=3–5. *$P$<0.05, **$P$<0.01, ****$P$<0.0001; n.s., not significant. CB, Coomassie Blue.

## Plasmids and siRNA

In the mammalian expression system, untagged Arl4D WT, Arl4A Q80L and T35N were cloned into the pSG5 vector (Stratagene) as described previously (Chen et al., 2020). HA-tagged Arl4s were cloned by fusing HA tag sequence at the C-terminus of *Arl4* in pSG5 vector. For human recombinant Arl4D protein induction in *Escherichia coli* (*E. coli*), the open reading frame (ORF) of Arl4D was cloned into pETDuet-1 vector with a His tag (Novagen). For expressing human Arl1, Arl4A and Arl4D proteins in yeast for yeast two-hybrid screening, ORFs of

human *Arl1*, *Arl4A* and *Arl4D* and their mutants were cloned into pBTM116 vector (Clontech Laboratories) with a LexA tag. For expressing human TBC1D15 protein in yeast, for yeast two-hybrid screening, the fragment 303–675 amino acids of human *TBC1D15* was cloned into pACT2 vector (Clontech Laboratories) with a HA tag. For expressing human TBC1D15 in mammalian cells, the ORF of human *TBC1D15* isoform 3 was amplified from human fetal brain cDNA library by PCR, fused with a Myc tag sequence at the N-terminus, and cloned into pSG5 vector. The primer sequence for PCR were as followers: Forward:

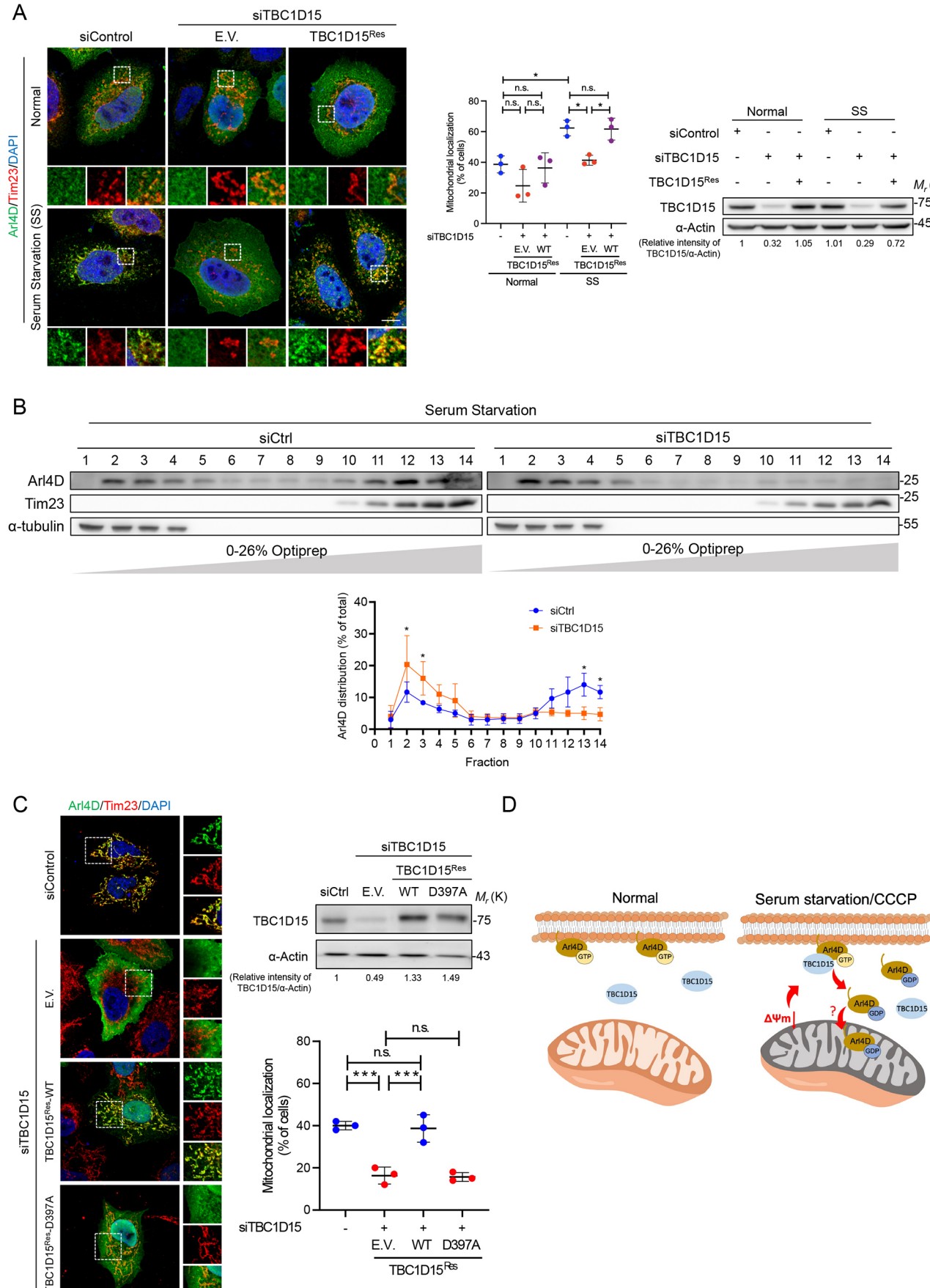

**Fig. 4.** See next page for legend.

**Fig. 4. TBC1D15 controls the translocation of Arl4D into mitochondria.**
(A) HeLa cells were treated with indicated siRNA and plasmids and incubation in the indicated medium for 24 h. Cells were then fixed and stained with anti-Arl4D (green) and anti-Tim23 (red) antibodies, and DAPI (blue). (B) Lysates from HeLa cells with TBC1D15 depletion by siRNA with 24-h serum starvation were subjected to subcellular fractionation to assess mitochondrial translocation of endogenous Arl4D. The percentage of total protein in each fraction is quantified (mean±s.d., $n$=3). An unpaired two-tailed $t$-test was applied to determine $P$-value. (C) HeLa cells treated with siControl or siTBC1D15 were transfected with Arl4D WT and rescued with indicated TBC1D15 plasmids before 24 h of serum starvation. After fixing, cells were stained with anti-Arl4D (green) and anti-Tim23 (red) antibodies, and DAPI (blue). Quantification of Arl4D localized at the mitochondria in A and C were calculated by determining the percentage of cells with colocalized Arl4D and Tim23 and is presented as mean±s.d. ($n$=3; 300 cells per group) in scatter plots. Knockdown efficiency of TBC1D15 was verified by immunoblotting. One-way ANOVA with Tukey's post hoc multiple comparison test was applied to determine $P$-value. *$P$<0.05, ***$P$<0.001. Scale bar: 10 µm. E.V., empty vector; SS, serum starvation medium. (D) Model of Arl4D translocation to mitochondria under the control of its GAP TBC1D15. Under normal conditions, activated GTP-bound Arl4D is located at the plasma membrane. Upon serum deprivation or CCCP treatment, TBC1D15 accesses to the plasma membrane to promote GTP-hydrolysis of Arl4D, which triggers the release of GDP-bound form of Arl4D from the plasma membrane and translocated to the mitochondria.

5′-ATAGGATCCATGGAACAAAAACTCATCTCAGAAGAG-GATCT-GATGGCGGCGGCGGGTG-3′; Reverse: 5′-TATAGATCTTCATGCAGG-TGTTAATCTGC-AGACATCTGAGGACACTGGTATCTGTGTTGG-3′. The TBC1D15-D397A mutant (loss of GAP activity) and the TBC1D15 siRNA-resistant TBC1D15 constructs were generated by PCR-mediated site-directed mutagenesis. The primer sequence used were as followers: TBC1D15[Res]: Forward 1: 5′-CTGATTGAATCTCTTGAAAAATATG-TCGTTCTGTGCGAATCTCCACAGGATAAAAGAAC-3′; Reverse 1: 5′-GTTCTTTTATCCTGTGGAGATTCGCACAGAACGACATATTTTTCA-AGAGATTCAATCAG-3′; Forward 2: 5′-AAAAATATGTCGTTCTGT-GCGAGTCACCTCAAGATAAAAGAACACTTCTTGTGAATTGTC-3′; Reverse 2: 5′-GACAATTCACAAGAAGTGTTCTTTTATCTTGAGGT-GACTCGCACAGAACGACATATTTTT-3′. TBC1D15-D397A: Forward: 5′-GAAGTCTTATCGAAAAAGCTGTTAACAGAACAGATCG-3′; Reverse: 5′-CGATCTGTTCTGTTAACAGCTTTTTCGATAAGACTTC-3′. For human Fis1 expression in mammalian cells, the ORF of human Fis1 was amplified by PCR from a brain cDNA library and cloned into pcDNA 3.0 vector (Invitrogen) with a HA tag. The primer sequences for PCR were: Forward: 5′-ATATAGAATTCATGGAGGCCGTGCTGAACGAG-3′; Reverse 5′-TATTCTAGATCAGGATTTGGACTTGGACACAGC-3′. For human recombinant TBC1D15 protein induction in *E. coli*, different fragments of human *TBC1D15* were cloned into pGEX-4T-1 vector (Clontech Laboratories) with a GST tag. For human recombinant Pak1-PBD induction in *E. coli*, the PBD fragment of human Pak1 was cloned into pGEX-4T-1 vector as previously described (Chen et al., 2020). Mito-QC reporter pcDNA3.1-2xcox8-GFP-mCherry and Tom20-mScarlet3 plasmids (as marker for mitochondrial outer membrane) were generously provided from Dr Wei-Chung Chiang (Institute of Biochemistry and Molecular Biology, School of Life Science, National Yang-Ming University, Taipei, Taiwan) and Dr Ya-Wen Liu (Institute of Molecular Medicine, College of Medicine, National Taiwan University, Taipei, Taiwan), respectively.

The sequences of the siRNAs used in this study were as follows: Control, 5′-UGGUUUACAUGUCGACUAAUU-3′; Arl4D, 5′-GGGAACCACUU-GACUGAGAUUU-3′; TBC1D15, 5′-GGUAUUGUGUGAAUCUCCA-CAGGAU-3′; and Rab7, 5′-CUAGAUAGCUGGAGAGAUGUU-3′.

### Antibodies
The primary antibodies used in this study were listed below: anti-α-tubulin (1:50,000, T5168, Sigma-Aldrich, as an internal control), anti-HA [1:5000, #901515, BioLegend for immunoblotting; 1:200, A190-108A, Bethyl Laboratories for immunofluorescence (IF) staining], anti-His (1:5000, #631212, Takara), anti-Myc (1:3000, 2278S, Cell Signaling Technology), anti-TBC1D15 (1:5000, GTX121081, GeneTex), anti-Tim23 (1:1000 for

immunoblotting; 1:100 for IF staining, #611223, BD Biosciences, as marker of mitochondrial inner membrane), anti-GAPDH (1:5000, GTX627408, GeneTex, as an internal control), anti-α-actin (1:5000, sc-58669, Santa Cruz Biotechnology, serving as marker for internal control), anti-Na⁺/K⁺ ATPase (1:1000, 3010S, Cell Signaling Technology). Antibodies against Arl4A, Arl4C or Arl4D were prepared as previously described (Chen et al., 2020; Li et al., 2007; Lin et al., 2022) and applied with a dilution ratio of 1:3000 for immunoblotting and 1:200 for IF staining. The secondary antibodies used in this study were: goat HRP-conjugated anti-rabbit and anti-mouse immunoglobulin antibodies (1:5000, NA934V/NA931V, GE Healthcare), Alexa Fluor 488-, 594- or 647-conjugated anti-rabbit and anti-mouse immunoglobulin secondary antibodies (1:1000, A-11034/A-11032/A-31573, Invitrogen). DAPI was purchased from Millipore (1:5000, S7113).

### Immunofluorescence staining and confocal microscopy
For immunofluorescence staining, cells were washed with PBS twice and fixed with 4% paraformaldehyde in PBS at 37°C for 15 min. After fixing, the cells were washed with PBS for 10 min three times and permeabilized with 0.1% Triton X-100 in PBS at room temperature (RT) for 5 min. The cells were then washed with PBS for 5 min three times, immersed in IF blocking solution [1% bovine serum albumin (BSA) in PBS], and placed on a shaker at RT for 30 min. After BSA blocking, they were incubated with primary antibodies in PBS containing 1% BSA for 21 h or at 4°C overnight (for anti-Tim23 antibody). After washing with PBS for 10 min three times, the cells were incubated with secondary antibody in PBS with 1% BSA for 1 h. After washing with PBS for 10 min three times, the coverslips were mounted on slides with mounting solution (Mowiol 4-88/DABCO). For confocal microscopy, the samples were imaged and analyzed using Carl Zeiss LSM 700 laser scanning confocal microscope with a 63×/1.4 NA oil objective lens. To quantify the colocalized signals, Zeiss ZEN Microscope Software (ZEN) was utilized to obtain Pearson's correlation coefficient for each cell. FIJI (ImageJ2) was used for measurements of the circularity of each mitochondria and the quantification of membrane-to-cytosol signal ratios (as described by Chen et al., 2020).

### Hypoxia-induced mitophagy induction and assessment
To induce mitophagy, HeLa cells transfected with the mito-QC reporter were incubated in 0.5% O₂, 5% CO₂ at 37°C in an anaerobic incubator (SCItive Hypoxia Workstation, Ruskinn) for 24 h before being fixed and imaged with a Carl Zeiss LSM 700 confocal microscope. Mitophagy counts per cell were measured in FIJI/ImageJ by identifying mCherry-only mitochondrial puncta (GFP signal lost after lysosomal degradation). Briefly, mCherry channel particles were identified after subtraction of scaled GFP signal (GFP multiplied by 1.0–2.0 for channel calibration) and counted per cell.

### Subcellular fractionation
Subcellular fractionation was conducted by density gradient centrifugation of harvested cellular postnuclear supernatant (PNS) as described previously (Li et al., 2012). Obtained subcellular fractions were analyzed by immunoblotting.

### *In vitro* binding assay
For the expression of GST fusion proteins in *E. coli*, TBC1D15 and its deletion mutants were cloned into pGEX vectors (GE Healthcare). For production of the GST fusion protein, the indicated constructs were transformed into BL21 cells. Colonies expressing recombinant proteins were grown at 37°C in Luria-Bertani medium. Protein was induced by 0.5 mM isopropyl β-D-1-thiogalactopyranoside (IPTG) for 3 h at 37°C. The cell pellet was incubated with lysozyme, sonicated, and centrifuged at 14,000 $g$ for 20 min to separate the soluble fraction (supernatant) and insoluble fraction (pellet). The supernatant was incubated with glutathione–Sepharose 4B beads (Amersham Pharmacia Biotech) at 4°C for 3 h. The beads were washed five times with cold PBS and then quantified by SDS-PAGE. Induction and purification of His- or GST-tagged fusion proteins was performed as previously described (Lin et al., 2020). The purified GST-tagged proteins were incubated with His-tagged Arl4A in 1 ml of binding buffer [20 mM Tris-HCl pH 8.0, 100 mM NaCl, 5 mM MgCl₂, 1 mM EDTA, 1 mM DTT, 250 mM sucrose, 10% glycerol, 0.1% Triton X-100 and

Journal of Cell Science

protease inhibitor cocktail (#04693116001, Roche)] on an end-over-end rotator at 4°C for 1 h. After 1 h, the GST beads were washed three times with 1 ml of wash buffer (*in vitro* binding buffer containing 1% Triton X-100). After removing the *in vitro* wash buffer completely, the beads were suspended in 10 µl 1× sample buffer for SDS-PAGE and immunoblotting analysis.

### Active Arl4D pulldown assay
HeLa cells transfected with Arl4D plasmid DNA were lysed in pull-down buffer (50 mM Tris-HCl pH 7.5, 150 mM NaCl, 5 mM MgCl₂, 0.5% NP-40, PI) on a shaker at 4°C for 10 min. The supernatants of the cell lysates were collected by centrifugation at 15,000 *g* for 15 min at 4°C, and incubated with 10 µg of GST or GST-tagged proteins bound to glutathione–Sepharose beads on an end-over-end rotator at 4°C for 2 h. After protein binding, the beads were washed with pulldown buffer containing 1.5% NP-40 on an end-over-end rotator at 4°C for 10 min five times before subjected to immunoblotting for analysis. Analysis of binding affinity was assessed by determining the value for the Arl4D input signals divided by amount of GST–Pak–PBD following densitometric quantification of bands.

### Co-immunoprecipitation assay
HeLa cells were lysed in lysis buffer (50 mM Tris-HCl, pH7.5, 150 mM NaCl, 0.5% NP-40 and protease inhibitor cocktail) at 4°C for 30 min. The lysates were clarified by centrifugation at 15,000 *g* for 30 min. The cell lysates were incubated with Myc-Trap (ChromoTek) at 4°C 15-30 min with rotation. Coimmunoprecipitated proteins were analyzed by immunoblotting. The procedure for endogenous co-IP was as described previously (Chang et al., 2025). Briefly, HeLa cells were subjected to a 2-h cross-linking with 100 mM dithiobis-[succinimidyl propionate] (DSP, Themo Fisher Scientific), followed by quenching, washing and lysis with 1 ml of lysis buffer (50 mM Tris-HCl pH 7.4, 150 mM NaCl, 1% NP-40, 1 mM EDTA, 5% glycerol, 0.1% Tween-20 and protease inhibitor cocktail). The supernatant was then incubated with anti-rabbit-IgG antibody (1:50, Cell Signaling) or anti-TBC1D15 (1:50, GeneTex) on an end-to-end rotator at 4°C for 2 h followed by incubation with Protein G magnetic beads (Merck Millipore) for another 2 h. The samples were analyzed by immunoblotting.

### Protein quantification and immunoblotting
Protein concentration in cell lysates was determined with a Bio-Rad DC protein assay (Bio-Rad) according to the manufacturer's protocols. Proteins were analyzed with separation by SDS-PAGE followed by transfer to PVDF membranes (Millipore). The transferred membrane was incubated with indicated antibodies and developed with an ECL system (Amersham-Pharmacia Biotech) as described previously (Chen et al., 2020; Lin et al., 2022, 2023). Uncropped image of blots from this study are shown in Fig. S5.

### Yeast two-hybrid screening
The yeast strain, L40 [MATα leu2 his3 trp1 LYS2::(lexAop) 4 HIS3 URA3:: (lexAop) 8 lacZ], was used to identify protein–protein interactions by both histidine auxotrophy and β-galactosidase expression. Yeast two-hybrid assays were performed as described previously (Li et al., 2007). Briefly, the yeast strain L40 was engineered with two interaction readouts, histidine auxotrophy and β-galactosidase expression, using the LexA DNA-binding domain and GAL4 activation domain system. A lithium acetate transformation protocol was used to screen a human fetal liver pACT2 cDNA library (Clontech) with ARL4D(Q80L) as bait. From screening 6×10⁶ clones for histidine auxotrophy and β-galactosidase activity, 33 clones were found to interact specifically with ARL4D-Q80L.

### GAP assay
GAP-accelerated GTP hydrolysis was measured using the EnzChek phosphate assay kit (Invitrogen). To obtain GTP-Arl4D proteins, Arl4D was incubated with a 25-fold molar excess of GTP on ice for 3 h. After incubation, the excess GTP was removed using an Amicon® ultra centrifugal filter. The GAP reaction mixture, containing 20 µM Arl4D-GTP complexes, was then mixed with 5 µM TBC1D15 303–675 (GAP). This mixture was loaded into 96-well plates, with each well containing a reaction buffer composed of 50 mM Tris-HCl pH 7.5, 150 mM NaCl,

10 mM MgCl₂, 200 µM 2-amino-6-mercapto-7-methylpurine riboside (MESG) and 1 U/ml of purine nucleoside phosphorylase (PNP). Phosphate (Pi) production was monitored by recording the change in absorbance at 360 nm using a spectrophotometer, with absorbance measurements taken every 5 s for up to 10 min. The $K_{obs}$ (rate constant) was calculated with the following formula: $K_{ob} = -\frac{1}{600}\ln\left(1 - \frac{F(t)-F_0}{F_\infty-F_0}\right)$.

### Statistical analysis
Statistical comparisons between treatments were performed by unpaired two-tailed parametric *t*-tests (Student's *t*-tests) or one-way analysis of variance (ANOVA) with Tukey's post hoc in GraphPad Prism 8. For the post hoc test, we used Tukey's method to compare all possible group pairings. Significant differences are indicated in the figure (*$P<0.05$; **$P<0.01$; ***$P<0.001$; ****$P<0.0001$).

### Acknowledgements
We thank Drs Randy Haun, Chia-Jung Yu, Ming-Chieh Lin, and Ya-Wen Liu for their critical review of this manuscript prior to submission. We also thank Dr You-An Su and Yen-Jung Hsu for their assistance in the quantification of mitochondria fragmentation with FIJI (ImageJ2) software. Finally, we thank the imaging core and biomedical resource center at the First Core Labs, National Taiwan University Medicine College for technical assistance.

### Competing interests
The authors declare no competing or financial interests.

### Author contributions
Conceptualization: F.-J.S.L.; Data curation: C.-T.C., T.-J.L., S.-J.L., T.-W.C.; Formal analysis: C.-T.C., S.-J.L., T.-W.C.; Funding acquisition: F.-J.S.L.; Investigation: F.-J.S.L.; Methodology: C.-T.C., T.-J.L., S.-J.L., T.-W.C.; Project administration: F.-J.S.L.; Resources: C.-T.C., T.-J.L., S.-J.L., T.-W.C.; Supervision: F.-J.S.L.; Validation: C.-T.C., T.-J.L., S.-J.L., T.-W.C., F.-J.S.L.; Visualization: C.-T.C., T.-J.L., T.-W.C.; Writing – original draft: C.-T.C., T.-J.L., S.-J.L., F.-J.S.L.; Writing – review & editing: C.-T.C., T.-W.C., F.-J.S.L.

### Funding
This work was supported by grants from the National Health Research Institutes (NHRI) of Taiwan (NHRI-EX113-11306BI), the Ministry of Science and Technology in Taiwan (NSTC 112-2311-B-002-012), and the Center of Precision Medicine from the Ministry of Education in Taiwan awarded to F.-J.S.L. Open Access funding provided by National Taiwan University. Deposited in PMC for immediate release.

### Data and resource availability
All relevant data and details of resources can be found within the article and its supplementary information.

### Peer review history
The peer review history is available online at https://journals.biologists.com/jcs/lookup/doi/10.1242/jcs.264304.reviewer-comments.pdf

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
