## [Peer Review File · Journal of Cell Science]

TBC1D15 functions as an Arl4D GAP and promotes mitochondrial translocation of Arl4D for organelle homeostasis

Chia-Tang Chen, Tsai-Jung Liu, Shin-Jin Lin, Ting-Wei Chang and Fang-Jen. S. Lee
DOI: 10.1242/jcs.264304

Editor: Michael Way

Review timeline

Original submission:	14 July 2025
Editorial decision:	28 August 2025
First revision received:	2 December 2025
Editorial decision:	30 December 2025
Second revision received:	4 February 2026
Accepted:	4 February 2026

Original submission

First decision letter

MS ID#: jcs.264304

MS TITLE: TBC1D15 functions as an Arl4D GAP and promotes mitochondrial translocation of Arl4D for organelle homeostasis

AUTHORS: Tsai-Jung Liu; Shin-Jin Lin; Chia-Tang Chen; Ting-Wei Chang; Fang-Jen S Lee

ARTICLE TYPE: Short Report

Dear Dr LEE,

It has taken longer than I would like because of the difficulty of finding reviewers during the summer, but we have now reached a decision on the above manuscript.

To see the reviewers' reports and a copy of this decision letter, please go to: View Reviewer Comments

As you will see, the reviewers raise a number of substantial criticisms that prevent me from accepting the paper at this stage. They, however, feel the study is interesting and suggest that a revised version might prove acceptable, if you can address their concerns. If you think that you can deal satisfactorily with the criticisms on revision, I would be pleased to see a revised manuscript. We would then return it to the reviewers.

Reviewer 1

Advance summary and potential significance to field

Liu et al. identify TBC1D15 as a GAP for ARL4D and propose that ARL4D inactivation by TBC1D15 enables ARL4D association with mitochondria. ARL4D was found to translocate to the mitochondria under conditions of serum starvation as well as mitochondrial depolarization by CCCP, and knockdown of TBC1D15 prevents ARL4D inactivation and translocation to mitochondria in response to serum starvation. Authors propose that TBC1D15 inactivation of ARL4D occurs at the plasma

membrane, and that mitochondrial ARL4D promotes hypoxia-induced mitophagy. The authors conclude that TBC1D15 and GDP bound ARL4D promote hypoxia induced mitophagy via an unknown pathway not requiring PINK1-Parkin.

The identification of TBC1D15 as a GAP for ARL4D potentially links TBC1D15 functions in mitochondrial biology to ARL4D, which is an interesting and important development both for mitochondrial biology as well as ARL biology. However, many important questions remain, such as how ARL4D is imported to the mitochondrial inner membrane (especially in cases where there is a lack of membrane potential), where this inactivation occurs, and how ARL4D may regulate mitophagy.

Major Criticisms

1. Authors conclude that TBC1D15 is somehow sensing mitochondrial stress or depolarization and, in response, inactivating ARL4D at the plasma membrane. This conclusion is not fully supported by the available evidence. The microscopy data (Fig. S3) is not compelling, authors would need to show subcellular fractionation to strengthen this claim.
2. Rather than some unknown signal going from stressed mitochondria to TBC1D15 at the plasma membrane, a more straightforward model would be that TBC1D15 is directly recruited to stressed mitochondria, where it inactivates ARL4D to enable its import. Yamano et al. 2014, 2017 have shown that TBC1D15 can be recruited to mitochondria, so it is unclear why authors did not test this possibility. Authors argue that the results of Yamano et al. 2017 are not relevant to their study because RAB7 acts downstream of Parkin, which is not present in HeLa cells (would their results differ in another cell type with Parkin?). However, Yamano 2017 report that TBC1D15 is recruited to mitochondria by Fis1, and not by Parkin-induced RAB cycling on mitochondria. Therefore, authors should assay 1) whether TBC1D15 is recruited to mitochondria in their experiments, 2) whether depletion of Fis1 prevents TBC1D15-mediated inactivation of ARL4D.
3. The effects upon ARL4D resulting from knockdown of TBC1D15 could be caused by loss of its GAP activity for RAB7 rather than ARL4D. The persistence of the reported cellular phenotypes in the absence of RAB7 activity is an important control that should be included to strengthen the findings of this manuscript.
4. The relationship between ARL4D import and mitochondrial membrane potential is unclear. Authors propose that mitochondrial depolarization is the driving factor for ARL4D import into mitochondria. This explanation may unify their data that mitochondrial ARL4D increases in response to CCCP and serum starvation, but not in response to glucose deprivation. However, no data is shown that serum starvation diminishes mitochondrial membrane potential. Furthermore, authors have previously published that ARL4D[T35N] depolarizes mitochondria upon import, so there is a chicken and egg problem - is depolarization upstream of ARL4D-GDP, or is it the other way around? Finally, how ARL4D is imported in response to mitochondrial depolarization is unclear. Authors have previously published that ARL4D resides inside of mitochondria; however, import into mitochondria is dependent upon membrane potential. When ARL4D association with mitochondria increases in response to CCCP, is ARL4D actually inside of mitochondria (i.e., proteinase k resistant)? Or is ARL4D stabilized on the outer mitochondrial membrane of depolarized mitochondria (analogous to PINK1)?
5. The authors claim that mitochondrial ARL4D mediates hypoxia-induced mitophagy; however, this is an overstatement based upon the data provided. ARL4D depletion modestly inhibits hypoxia-induced mitophagy only when the additional stressor of serum starvation is layered on, and whether ARL4D is imported into mitochondria under hypoxia was not tested. Though serum starvation also promotes ARL4D association with mitochondria, it is not correlated with an increase in mitophagy (Fig. S2). In addition, the authors have previously reported that ARL4D[T35N] import into mitochondria dissipates membrane potential, so any mitophagy induced by ARL4D could be secondary to a loss in membrane potential. Therefore, the function of mitochondrial ARL4D is unclear. Are there changes to other mitochondrial parameters when ARL4D is imported, such as ATP production, mtROS, etc.? Are there other mitochondrial stress pathways enhanced by ARL4D? Does ARL4D change in response to other mitochondrial effects known to be linked to TBC1D15?

Minor Criticisms

1. TBC1D15 is identified as a potential ARL4D GAP via a yeast two-hybrid assay, but no data from this assay is shown in the paper.
2. No scale bar is shown in Fig. 1B.
3. Fig. 1D, it's unclear which lanes on the gel were quantified.

4. 1F, labels on graph should match labels on blot (dTBC should be labeled as d326-562, or vice versa).
5. 2A,B: authors should quantify GTP hydrolysis rate and show replicates.
6. The blots in Fig. 4 should be quantified.
7. Page 5, Line 54: Discussion of the arginine finger in ELMOD2 should cite the following work: East et al. 2012, "ELMO domains, evolutionary and functional characterization of a novel GTPase-activating protein (GAP) domain for Arf protein family GTPases."
8. All siRNA experiments need to be controlled for off target effects using either 1) rescue or 2) inclusion of an additional, sequence-independent siRNAs.
9. The total number of cells quantified per condition needs to be listed in all figure legends.
10. Fig. S2: it is unclear what "mitophagy counts" means, this information should be added to the methods and the figure legend. It would also be helpful to include an inset in each image should an example of a "mitophagy count."
11. Many microscopy images lack quantification, such as Figures S1 C-E and S4, all microscopy data in the manuscript should be quantified.
12. Figure S4 does not agree with other data shown in the paper (siTBC1D15 increases mitochondrial ARL4D while CCCP does not?), was the data mislabeled?
13. Microscopy images showing overexpressed mutant ARL4D should include controls with overexpressed wild type protein. Figure S2C should include an untreated control to facilitate a clearer comparison of ARL4D's subcellular localization.
14. Demonstrating some of the key findings in a different cell line would provide stronger support for the paper's argument, especially in light of the fact that HeLa cells lack Parkin.
15. Page 4, Line 51, citations should be provided characterizing the myristoylation-deficient ARL4D mutant.
16. Page 5, Line 21, replace "in vivo" with "in cultured cells."

Reviewer 2

Advance summary and potential significance to field

This paper by Liu et al. reports the identification of TBCD15 as a GTPase-activating protein (GAP) for ARL4D. The initial section of the manuscript reinforces previous findings from the Lee lab, demonstrating that GDP-bound ARL4D localizes to mitochondria. The authors then present results from a yeast two-hybrid (Y2H) screen, which identified TBCD15 as a potential interactor of both wild-type and constitutively active ARL4D. Given that TBCD15 is known to function as a GAP for Rab7, the authors hypothesize that it may similarly act on ARL4D. Supporting this, they provide both in vitro and cell-based assays indicating that TBCD15 promotes GTP hydrolysis of ARL4D. Knockdown of TBCD15 via siRNA led to increased GTP-loading of ARL4D. Consistent with the model that only GDP-bound ARL4D localizes to mitochondria, TBCD15 depletion resulted in ARL4D being excluded from mitochondrial compartments.

Overall, this is a compelling study that proposes a potentially significant finding: Rab GAPs may also regulate atypical Arf-related proteins such as ARL4D. While the core data are solid, several additional experiments could further substantiate the conclusions.

Comments for the author

Major comments

1. Endogenous ARL4D. All experiments rely on overexpression of ARL4D. It would strengthen the study if the authors could tag endogenous ARL4D (e.g., with HA or FLAG) using CRISPR and perform two key experiments:

- (1) Demonstrate interaction between endogenous ARL4D and TBCD15.
- (2) Assess whether TBCD15 depletion affects mitochondrial localization of endogenous ARL4D.

2. Figure 1 Presentation and Interpretation In Figure 1, the cell fractionation data should be consolidated into a single panel to facilitate comparison across conditions (control, high cell density, and serum starvation). Specifically, panel 1e should be integrated with 1a. Additionally,

the authors should address the discrepancy between biochemical co-fractionation with mitochondrial markers under both conditions, and the observed mitochondrial relocalization only during starvation.

3. Yeast Two-Hybrid Dataset Transparency In current cell biology standards, full datasets from screens (e.g., siRNA, CRISPR, proteomics) are typically presented. The authors should include the complete Y2H dataset, allowing readers to evaluate other potential interactors, including known effectors or positive controls.

4. Figure 2H Clarity. The data in Figure 2H are not convincing due to high background. The authors should consider alternative approaches or replicate the experiment with improved signal-to-noise ratio.

5. ARL4D Biochemical Properties The manuscript lacks clarity on the biochemical behaviour of ARL4D. If ARL4D functions as a typical GTPase, only a small fraction should be GTP-bound at steady state. This raises the question: should ARL4D consistently localize to mitochondria, with localization decreasing upon GTP-loading? Or is ARL4D atypical, predominantly GTP-bound, requiring GAP activity to generate detectable GDP-bound species? Clear statements on these points would greatly enhance the conceptual framework of the study.

6. Structural Compatibility of ARL4D and Rab GAPs Given the structural differences between ARFs and Rabs—particularly the N-terminal helical extension in ARFs—it is difficult to envision how ARL4D would interact with a Rab GAP domain. Structural modeling using AlphaFold to predict interactions between ARL4D, Rab7 (positive control), and the GAP domain of TBCD15 would provide valuable insight. Evidence supporting the structural feasibility of such a complex would significantly strengthen the manuscript. Future work could aim to obtain empirical structural data.

Minor Comments

1. Experimental Detail and Accessibility. Although this is a short report, the experimental descriptions are minimal and may be challenging for non-specialists. For broader accessibility (e.g., JCS readership), the authors should expand on the experimental details in the main text. For example, in Figure 1a, the role of the mitochondrial marker Tim23 is not explained, making interpretation difficult.

2. Terminology in Figure 3 In the description of Figure 3b, the term "nullified" is too strong given the partial nature of the effect. "Partially restores" would be more appropriate. Similarly, in panel 3d, "partially restored" better reflects the data.

First revision

Author response to reviewers' comments

Replies to Reviewers

Reviewer #1:

Liu et al. identify TBC1D15 as a GAP for ARL4D and propose that ARL4D inactivation by TBC1D15 enables ARL4D association with mitochondria. ARL4D was found to translocate to the mitochondria under conditions of serum starvation as well as mitochondrial depolarization by CCCP, and knockdown of TBC1D15 prevents ARL4D inactivation and translocation to mitochondria in response to serum starvation. Authors propose that TBC1D15 inactivation of ARL4D occurs at the plasma membrane, and that mitochondrial ARL4D promotes hypoxia-induced mitophagy. The authors conclude that TBC1D15 and GDP bound ARL4D promote hypoxia induced mitophagy via an unknown pathway not requiring PINK1-Parkin.

The identification of TBC1D15 as a GAP for ARL4D potentially links TBC1D15 functions in mitochondrial biology to ARL4D, which is an interesting and important development both for mitochondrial biology as well as ARL biology. However, many important questions remain, such as how ARL4D is imported to the mitochondrial inner membrane (especially in cases where there is a lack of membrane potential), where this inactivation occurs, and how ARL4D may regulate mitophagy.

Major Criticisms

1. Authors conclude that TBC1D15 is somehow sensing mitochondrial stress or depolarization and, in response, inactivating ARL4D at the plasma membrane. This conclusion is not fully supported by the available evidence. The microscopy data (Fig. S3) is not compelling, authors would need to show subcellular fractionation to strengthen this claim.

Response: Thank you for your insightful feedback. In response, we have conducted subcellular fractionation in HeLa cells to further investigate this point. The fractionation results indicate that upon TBC1D15 depletion by siRNA, there is a noticeable redistribution of Arl4D from the cytosol fraction to the plasma membrane. This change supports the observation of increased membrane-associated Arl4D when TBC1D15 level is reduced. We have included these findings in a new Figure S3C to provide additional clarity.

2. Rather than some unknown signal going from stressed mitochondria to TBC1D15 at the plasma membrane, a more straightforward model would be that TBC1D15 is directly recruited to stressed mitochondria, where it inactivates ARL4D to enable its import. Yamano et al. 2014, 2017 have shown that TBC1D15 can be recruited to mitochondria, so it is unclear why authors did not test this possibility. Authors argue that the results of Yamano et al. 2017 are not relevant to their study because RAB7 acts downstream of Parkin, which is not present in HeLa cells (would their results differ in another cell type with Parkin?). However, Yamano 2017 report that TBC1D15 is recruited to mitochondria by Fis1, and not by Parkin-induced RAB cycling on mitochondria. Therefore, authors should assay 1) whether TBC1D15 is recruited to mitochondria in their experiments, 2) whether depletion of Fis1 prevents TBC1D15-mediated inactivation of ARL4D.

Response: We appreciate the thoughtful comment on our manuscript. We have performed experiments to answer the reviewer's questions. First, to examine whether the recruitment of TBC1D15 to the mitochondria is required for Arl4D mitochondria translocation, we performed staining against Myc-tagged TBC1D15 expressed in HeLa cells under serum starvation and CCCP treatment (conditions that drive Arl4D translocation). We found that TBC1D15 remained cytosolic under both treatments, indicating no detectable stress-induced recruitment of TBC1D15 to mitochondria in our system. Second, to rule out the possibility that the lack of TBC1D15 translocation is caused by defects in our cell culture system and procedural problems, we co-transfected HA-tagged Fis1 protein with Myc-tagged TBC1D15 and observed that TBC1D15 was successfully recruited to the mitochondria, reproducing published Fis1-dependent recruitment and ruling out technical failure. Finally, we co-expressed Arl4D with or without HA-tagged Fis1 and Myc-tagged TBC1D15 to assess whether translocation of Arl4D to the mitochondria is regulated by Fis1-mediated mitochondrial recruitment of TBC1D15. We found that recruitment of TBC1D15 to the mitochondria by Fis1 did not alter the fraction of mitochondria-localized Arl4D compared with controls in HeLa cells. These results were quantified across independent replicates, as shown in the figure below. Together these data show that, while TBC1D15 can be recruited to mitochondria by Fis1, Fis1-mediated mitochondrial recruitment of TBC1D15 does not influence Arl4D

translocation in our HeLa-cell assays, making Fis1-dependent TBC1D15 mitochondrial targeting unlikely to be the regulatory mechanism for Arl4D in this context.

3. The effects upon ARL4D resulting from knockdown of TBC1D15 could be caused by loss of its GAP activity for RAB7 rather than ARL4D. The persistence of the reported cellular phenotypes in the absence of RAB7 activity is an important control that should be included to strengthen the findings of this manuscript.

Response: Thank you for pointing out the potential role of TBC1D15's GAP activity for RAB7 in our observed effects on ARL4D. In response, we conducted additional experiments to address this concern. As presented in Figure 4A, we initially demonstrated that TBC1D15 knockdown reduces the mitochondrial translocation of Arl4D under serum starvation. To explore the involvement of Rab7, given its role as a GAP substrate for TBC1D15, we performed experiments where HeLa cells were transfected with Rab7 siRNA. We assessed Arl4D mitochondrial translocation under both normal and serum-starved conditions. Our findings show that Rab7 depletion does not impair serum starvation-induced Arl4D translocation. These results suggest that Rab7 is unlikely to mediate the Arl4D translocation phenotype, thus supporting the specificity of our original findings.

4. The relationship between ARL4D import and mitochondrial membrane potential is unclear. Authors propose that mitochondrial depolarization is the driving factor for ARL4D import into mitochondria. This explanation may unify their data that mitochondrial ARL4D increases in response to CCCP and serum starvation, but not in response to glucose deprivation. However, no data is shown that serum starvation diminishes mitochondrial membrane potential. Furthermore, authors have previously published that ARL4D[T35N] depolarizes mitochondria upon import, so there is a chicken and egg problem - is depolarization upstream of ARL4D-GDP, or is it the other way around? Finally, how ARL4D is imported in response to mitochondrial depolarization is unclear. Authors have previously published that ARL4D resides inside of mitochondria; however, import into mitochondria is dependent upon membrane potential. When ARL4D association with mitochondria increases in response to CCCP, is ARL4D actually inside of mitochondria (i.e., proteinase K resistant)? Or is ARL4D stabilized on the outer mitochondrial membrane of depolarized mitochondria (analogous to PINK1)?

Response: We thank the reviewer for raising these important points. To address whether serum starvation affects $\Delta\psi_m$ we stained HeLa cells with MitoTracker Red and quantified fluorescence intensity: starvation reduces the MitoTracker signal in a manner similar to CCCP, consistent with published reports that starvation can lower mitochondrial membrane potential (Papucci et al., 2003). We note that our 2012 study (Li et al., 2012) showed that forced import of ARL4D[T35N] can depolarize mitochondria, but deletion of ARL4D's C-terminal region abolishes that depolarizing effect while still allowing translocation, indicating that ARL4D-induced depolarization is separable from translocation. Together these observations argue that loss of $\Delta\psi_m$ can accompany or amplify ARL4D mitochondrial accumulation but is not strictly required as the initiating trigger.

With respect to sub-mitochondrial localization after depolarizing treatments, our prior study (Li et al., 2012) used proteinase K protection assays and electron microscopy to demonstrate that the ARL4D-T35N GDP-mimetic form can be present within the mitochondrial matrix. While mitochondrial import of most proteins depends on membrane potential, Andreas Geissler et al. reported that certain proteins can undergo sorting in a $\Delta\psi$ -independent manner, where conformational flexibility enables formation under conditions of reduced proton motive force. Thus, although both CCCP and serum starvation decrease mitochondrial membrane potential, we suggest this decrease did not affect ARL4D sorting into mitochondria. However, we have not yet included new proteinase K protection assays in the present manuscript to definitively distinguish luminal import from stable association at the outer mitochondrial membrane following CCCP treatment. Therefore, in this manuscript we conservatively describe the phenomenon as increased mitochondrial translocation, while the exact mechanism facilitating ARL4D-GDP import into mitochondria under reduced membrane potential remains open for interpretation. Given that both CCCP and serum starvation induce ROS, and that ROS can regulate signaling between mitochondria and cytosolic effectors, we propose that ROS may function as an upstream cue that activates TBC1D15 at the plasma membrane, thereby facilitating ARL4D translocation (Samluk et al., 2019; Scherz-Shouval et al., 2007).

Reference:

Papucci, L., Schiavone, N., Witort, E., Donnini, M., Lapucci, A., Tempestini, A., ... & Capaccioli, S. (2003). Coenzyme q10 prevents apoptosis by inhibiting mitochondrial depolarization independently of its free radical scavenging property. *Journal of Biological Chemistry*, 278(30), 28220-28228.

Li, C.-C., T.-S. Wu, C.-F. Huang, L.-T. Jang, Y.-T. Liu, S.-T. You, G.-G. Liou, and F.-J.S. Lee. 2012. GTP-Binding-Defective ARL4D Alters Mitochondrial Morphology and Membrane Potential. *PLOS ONE*. 7:e43552.

Geissler, A., Krimmer, T., Bomer, U., Guiard, B., Rassow, J., & Pfanner, N. (2000). Membrane potential-driven protein import into mitochondria: the sorting sequence of cytochrome b 2Modulates the dependence of translocation of the matrix-targeting sequence. *Molecular biology of the cell*, 11(11), 3977-3991.

Samluk, L., Urbanska, M., Kisieleska, K., Mohanraj, K., Kim, M. J., Machnicka, K., ... & Chacinska, (2019). Cytosolic translational responses differ under conditions of severe short-term and long-term mitochondrial stress. *Molecular biology of the cell*, 30(15), 1864-1877.

Scherz-Shouval, R., Shvets, E., Fass, E., Shorer, H., Gil, L., & Elazar, Z. (2007). Reactive oxygen species are essential for autophagy and specifically regulate the activity of Atg4. *The EMBO journal*, 26(7), 1749-1760.

5. The authors claim that mitochondrial ARL4D mediates hypoxia-induced mitophagy; however, this is an overstatement based upon the data provided. ARL4D depletion modestly inhibits hypoxia-induced mitophagy only when the additional stressor of serum starvation is layered on, and whether ARL4D is imported into mitochondria under hypoxia was not tested. Though serum starvation also promotes ARL4D association with mitochondria, it is not correlated with an increase in mitophagy (Fig. S2). In addition, the authors have previously reported that ARL4D[T35N] import into mitochondria dissipates membrane potential, so any mitophagy induced by ARL4D could be secondary to a loss in membrane potential. Therefore, the function of mitochondrial ARL4D is unclear. Are there changes to other mitochondrial parameters when ARL4D is imported, such as ATP production, mtROS, etc.? Are there other mitochondrial stress pathways enhanced by ARL4D? Does ARL4D change in response to other mitochondrial effects known to be linked to TBC1D15?

Response: We thank the reviewer for these important points and agree that our original wording overstated the conclusions. Overall, our experimental data and the statements in our main text agree with the conclusion that serum starvation or other events that cause Arl4D inactivation and mitochondrial translocation must occur in parallel for Arl4D to facilitate hypoxia-induced mitophagy. However, the original section title "Knockdown of Arl4D impairs mitophagy" may be misleading, overemphasizing the role of Arl4D with its translocation to the mitochondria. We have revised the section title in the manuscript to make our statement more precise as following:

“Mitochondria-residing Arl4D facilitates hypoxia-induced mitophagy.”

We acknowledge the chicken-and-egg issue with $\Delta\psi_m$: prior work shows forced ARL4D import can depolarize mitochondria. We, therefore, emphasize that ARL4D accumulation and loss of membrane potential may be interrelated and that, in our experiments, ARL4D accumulation under CCCP/serum starvation can occur in contexts where depolarization is present or amplified. We explicitly note that we cannot yet resolve whether $\Delta\psi_m$ is strictly upstream of ARL4D import in all conditions.

Because this is a short report (not to exceed 3000 words, including the main text and figure legends, and the total number of displayed items [Figures and Tables] must not exceed four), we cannot extend the TBC1D15-Arl4D findings to mitochondrial functional consequences here; however, we acknowledge that additional experiments will be required to define ARL4D's mitochondrial functions and mechanisms in future studies.

Minor Criticisms

1. TBC1D15 is identified as a potential ARL4D GAP via a yeast two-hybrid assay, but no data from this assay is shown in the paper.

Response: As suggested by the reviewer, we have listed our dataset of identified potential interactors for Arl4D from the yeast-two hybrid screening (new Figure 2A) as following. In addition, the yeast two-hybrid result is shown in Fig. 2C, which demonstrates TBC1D15's selective interaction with ARL4D among ARL4 isoforms and stronger binding to wild-type and constitutively-active ARL4D.

Human fetal liver cDNA
Protein name
Heterochromatin protein 1 α
Transforming acidic coiled-coil containing Huntingtin interacting protein K (HYPK)
α -catenin
Heterogeneous nuclear ribonucleoprotein A1 cyclic C (CCNC)
TBC1 domain family, member 15
Homeo box B5 (HOXB5)
alpha-1-antichymotrypsin (AACT)
gamma globin
hypothetical protein (HSPC031)
Farnesy pyrophosphate synthetase C-Nap1

2. No scale bar is shown in Fig. 1B.

Response: A scale bar has been added to the edited Fig. 1B in the revised manuscript.

3. Fig. 1D, it's unclear which lanes on the gel were quantified.

Response: The quantified lanes correspond to the Arl4D bands pulled down by GST-Pak-PBD in the active-Arl4D assay; this is now explicitly indicated in the methods for the Active Arl4D pull-down assay, as following:

“Analysis of binding affinity is assessed by Arl4D input signals divided by amount of GST-Pak-PBD following densitometric quantification of bands.”

4. 1F, labels on graph should match labels on blot (dTBC should be labeled as d326-562, or vice versa).

Response: Labels in Fig. 2H have been updated so the graph and blot use the same nomenclature (d326- 562).

5. 2A,B: authors should quantify GTP hydrolysis rate and show replicates.

Response: As suggested by the reviewer, we quantified GTP hydrolysis rates and present replicated data in Figures 3A and 3B (results are shown below).

6. The blots in Fig. 4 should be quantified.

Response: As suggested by the reviewer, the quantification of Figure 4 has been added.

7. Page 5, Line 54: Discussion of the arginine finger in ELMOD2 should cite the following work: East et al. 2012, "ELMO domains, evolutionary and functional characterization of a novel GTPase-activating protein (GAP) domain for Arf protein family GTPases."

Response: The citation was updated as recommended: "Arf GAP proteins exhibit a conserved structural motif known as the "arginine finger," which plays a crucial role in GTP hydrolysis (Donaldson and Jackson, 2011; East et al., 2012; Sztul et al., 2019)."

References:

East, M.P., J.B. Bowzard, J.B. Dacks, and R.A. Kahn. 2012. ELMO domains, evolutionary and functional characterization of a novel GTPase-activating protein (GAP) domain for Arf protein family GTPases. *Journal of Biological Chemistry*. 287:39538-39553.

8. All siRNA experiments need to be controlled for off target effects using either 1)

rescue or 2) inclusion of an additional, sequence-independent siRNAs.

Response: Thank you for your suggestion regarding the control of off-target effects in our siRNA experiments. We have addressed this by repeating all siRNA knockdown experiments (Figures 3D, 4A, S2B, S3B, and S4) with rescue controls. The updated figures now include panels that demonstrate the restoration of the target protein and the reversal of the knockdown phenotype, providing additional confidence in the specificity of our findings.

9. The total number of cells quantified per condition needs to be listed in all figure legends.

Response: We appreciate the feedback. Because of word limits in the short report, we display the sample size (n) on each statistical plot within the figures (Prism graphs).

10. Fig. S2: it is unclear what "mitophagy counts" means, this information should be added to the methods and the figure legend. It would also be helpful to include an inset in each image should an example of a "mitophagy count."

Response: We updated the Methods and Fig. S2 legend with a concise definition: "Mitophagy counts per cell were measured in FIJI/ImageJ by identifying mCherry-only mitochondrial puncta (GFP signal lost after lysosomal degradation). Briefly, mCherry channel particles were identified after subtraction of scaled GFP signal (GFP multiplied by 1.0-2.0 for channel calibration) and counted per cell." We also added inset panels in Fig. S2 showing representative examples of an annotated mitophagy count indicated by arrowheads, as shown below.

Fig. S2B

Fig. S2C

11. Many microscopy images lack quantification, such as Figures S1 C-E and S4, all microscopy data in the manuscript should be quantified.

Response: Thank you for highlighting the need for quantification in our microscopy images. We have now quantified all the indicated panels (Figures S1 C-E and S4). The methods used for quantification, including the calculation of Pearson's correlation coefficient for colocalization, are detailed in the Methods section. Additionally, the sample sizes (n) are provided in the corresponding figure statistical plots to ensure clarity and transparency.

12. Figure S4 does not agree with other data shown in the paper (siTBC1D15 increases mitochondrial ARL4D while CCCP does not?), was the data mislabeled?

Response: We thank the reviewer's comment for pointing out this error. The panel in Fig. S4 was incorrect; we have replaced it with replicate experiments that include rescue controls (as described in response to comment #8). The corrected Fig. S4 shows that TBC1D15 depletion inhibits

translocation of Arl4D to mitochondria, consistent with the rest of the dataset.

13. Microscopy images showing overexpressed mutant ARL4D should include controls with overexpressed wild type protein. Figure S2C should include an untreated control to facilitate a clearer comparison of ARL4D's subcellular localization.

Response: As suggested, we have added wild-type ARL4D controls to all experiments that used mutant constructs. In the revised Fig. S1E we show that CCCP induces mitochondrial translocation of wild-type ARL4D. Fig. S2C now includes an untreated (normoxia) control and wild-type ARL4D, demonstrating that (a) ARL4D does not translocate to mitochondria under hypoxia alone without serum starvation, and (b) wild-type ARL4D does not enhance mitophagy under these conditions, in contrast to the T35N mutant.

14. Demonstrating some of the key findings in a different cell line would provide stronger support for the paper's argument, especially in light of the fact that HeLa cells lack Parkin.

Response: We validated key observations in MDA-MB-231 cells (which express endogenous Parkin/PINK1). Under serum starvation and CCCP treatment, ARL4D translocated to mitochondria in MDA-MB-231 cells, supporting that the mitochondrial association of ARL4D occurs independently of HeLa-specific Parkin deficiency, as shown below.

15. Page 4, Line 51, citations should be provided characterizing the myristoylation-deficient ARL4D mutant.

Response: We have cited the ARL4D G2A (myristoylation-deficient) mutant and its characterization with reference to the following two papers:

Li CC, Wu TS, Huang CF, Jang LT, Liu YT, et al. (2012) GTP-Binding-Defective ARL4D Alters Mitochondrial Morphology and Membrane Potential. PLOS ONE 7(8): e43552.

Harroun, T. A., Bradshaw, J. P., Balali-Mood, K., & Katsaras, J. (2005). A structural study of the myristoylated N-terminus of ARF1. *Biochimica et Biophysica Acta (BBA)-Biomembranes*, 1668(1), 138- 144.

16. Page 5, Line 21, replace "in vivo" with "in cultured cells."

Response: As suggested, the sentence was corrected to: "Using GST-Pak1-PBD to pull down GTP-Arl4D, we found cells expressing TBCD15 WT rather than D397A mutant reduced the level of active Arl4D (Fig. 3C), showing that the regulation of GTP-bound Arl4D depends on the GAP activity of TBCD15 in cultured cells."

Reply to Reviewer #2

Reviewer #2 (summary of the advance made in this paper and its potential significance to the field):

This paper by Liu et al. reports the identification of TBCD15 as a GTPase-activating protein (GAP) for ARL4D. The initial section of the manuscript reinforces previous findings from the Lee lab, demonstrating that GDP-bound ARL4D localizes to mitochondria. The authors then present results from a yeast two-hybrid (Y2H) screen, which identified TBCD15 as a potential interactor of both wild-type and constitutively active ARL4D. Given that TBCD15 is known to function as a GAP for Rab7, the authors hypothesize that it may similarly act on ARL4D. Supporting this, they provide both in vitro and cell-based assays indicating that TBCD15 promotes GTP hydrolysis of ARL4D. Knockdown of TBCD15 via siRNA led to increased GTP-loading of ARL4D. Consistent with the model that only GDP-bound ARL4D localizes to mitochondria, TBCD15 depletion resulted in ARL4D being excluded from mitochondrial compartments.

Overall, this is a compelling study that proposes a potentially significant finding: Rab GAPs may also regulate atypical Arf-related proteins such as ARL4D. While the core data are solid, several additional experiments could further substantiate the conclusions.

Suggestions to Authors

Major Comments

1. Endogenous ARL4D. All experiments rely on overexpression of ARL4D. It would strengthen the study if the authors could tag endogenous ARL4D (e.g., with HA or FLAG) using CRISPR and perform two key experiments:

- (1) Demonstrate interaction between endogenous ARL4D and TBCD15.
- (2) Assess whether TBCD15 depletion affects mitochondrial localization of endogenous ARL4D.

Response: We agree that endogenous validation would strengthen the study. Generating a CRISPR knock-in in HeLa cells is technically challenging due to their complex karyotype and multiple ARL4D copies on chromosome 17 (commonly present in four copies). Therefore, instead of attempting a knock-in, we performed two orthogonal endogenous assays.

Reference:

Macville, M., Schröck, E., Padilla-Nash, H., Keck, C., Ghadimi, B. M., Zimonjic, D., ... & Ried, T. (1999). Comprehensive and definitive molecular cytogenetic characterization of HeLa cells by spectral karyotyping. *Cancer research*, 59(1), 141-150.

"Chromosome 17 was commonly present as four normal copies; therefore, CGH should display a copy number gain. However, only a slight increase in copy number gain was detected, suggesting that tetraploidy for this chromosome was not present in >50% of the cell population. Previous G-banding studies have usually identified three chromosomes 17."

- (1) Endogenous co-immunoprecipitation: using anti-TBC1D15 to IP endogenous TBC1D15 from HeLa lysates, we detected endogenous ARL4D in the TBC1D15 IP (new Fig. 2E), supporting an intrinsic ARL4D-TBC1D15 association.

- (2) Subcellular fractionation of endogenous ARL4D: biochemical fractionation of control versus siTBC1D15 HeLa cells under serum starvation shows that depletion of TBC1D15 reduces the amount of endogenous ARL4D in the mitochondrial fraction, consistent with TBC1D15 regulating endogenous ARL4D mitochondrial association (new Fig. 4B).

2. Figure 1 Presentation and Interpretation. In Figure 1, the cell fractionation data should be consolidated into a single panel to facilitate comparison across conditions (control, high cell density, and serum starvation). Specifically, panel 1e should be integrated with 1a. Additionally, the authors should address the discrepancy between biochemical co-fractionation with mitochondrial markers under both conditions, and the observed mitochondrial relocalization only during starvation.

Response: We appreciate this insightful suggestion. We deliberately presented the fractionation experiments in separate panels to (1) highlight the chronological discovery—initial observation during prolonged culture (96 h) followed by replication under a defined physiological perturbation (24 h serum starvation)—and (2) emphasize important mechanistic differences between the conditions. Prolonged culture (96 h) exposes cells to multiple, heterogeneous stresses that can produce partial biochemical co-fractionation of ARL4D with mitochondrial markers without uniform, microscopy-detectable mitochondrial relocalization. By contrast, acute serum starvation (24 h) is a more specific stimulus that yields both biochemical enrichment in mitochondrial fractions and clearer microscopy evidence of mitochondrial translocation in many cells. We have revised the Results and Discussion (shown below) to state this distinction succinctly and to clarify why the datasets remain shown separately rather than consolidated into a single panel.

“We found that Arl4D translocates to the mitochondria as cell culture duration increases up to 96 hours (Fig. 1A). Over this period, cell density and stress accumulate while glucose and serum growth factors levels in the medium decline.”

“Additionally, fractionation experiments revealed that endogenous Arl4D translocated to mitochondria during 24 hours of serum starvation, evidenced by a shift to heavier fractions that co-localized with Tim23 (Fig. 1E).”

3. Yeast Two-Hybrid Dataset Transparency. In current cell biology standards, full datasets from screens (e.g., siRNA, CRISPR, proteomics) are typically presented. The authors should include the complete Y2H dataset, allowing readers to evaluate other potential interactors, including known effectors or positive controls.

Response: We appreciate the reviewer's comment on this matter. To provide data transparency, we listed our complete dataset of identified possible interactors for Arl4D from the yeast-two hybrid screening assay in the newly added **Figure 2A**, as shown below.

Human fetal liver cDNA
Protein name
Heterochromatin protein 1 α
Transforming acidic coiled-coil containing Huntingtin interacting protein K (HYPK)
α -catenin
Heterogeneous nuclear ribonucleoprotein A1 cyclic C (CCNC)
TBC1 domain family, member 15 (TBC1D15)
Homeo box B5 (HOXB5)
alpha-1-antichymotrypsin (AACT)
gamma globin
hypothetical protein (HSPC031)
Farnesy pyrophosphate synthetase
C-Nap1

4. **Figure 2H Clarity.** The data in Figure 2H are not convincing due to high background. The authors should consider alternative approaches or replicate the experiment with improved signal-to-noise ratio.

Response: We agree the original panel had high background. We have replaced the problematic panel (previously labeled Fig. 2F/2H) with replicate experiments showing improved signal-to-noise. The new data (Fig. 2H in the revised manuscript) use optimized antibody blocking and longer washes, increased input signal with reduced exposure, and inclusion of negative-control lanes to reduce background and clarify specific bands.

5. **ARL4D Biochemical Properties.** The manuscript lacks clarity on the biochemical behaviour of ARL4D. If ARL4D functions as a typical GTPase, only a small fraction should be GTP-bound at steady state. This raises the question: should ARL4D consistently localize to mitochondria, with localization decreasing upon GTP-loading? Or is ARL4D atypical, predominantly GTP-bound, requiring GAP activity to generate detectable GDP-bound species? Clear statements on these points would greatly enhance the conceptual framework of the study.

Response: We appreciate this insightful feedback. Previous work have revealed that Arl4D was detected more significantly on the plasma membrane, and this plasma membrane localization is dependent on the myristoylation and GTP-binding of Arl4D (Li, Chiang et al. 2007). We have edited our manuscript to clarify this issue as following: "**While wild-type Arl4D was mostly found active and residing on the plasma membrane, our previous study has shown that the nucleotide-binding-defective Arl4D mutant (Arl4D-T35N) localizes to the mitochondria and causes mitochondrial depolarization and fragmentation (Jacobs et al., 1999; Li et al., 2007; Li et al., 2012).**"

Reference:

Li, C.-C., T.-C. Chiang, T.-S. Wu, G. Pacheco-Rodriguez, J. Moss and F.-J. S. Lee (2007). "ARL4D recruits cytohesin-2/ARNO to modulate actin remodeling." *Molecular biology of the cell* **18**(11):

4420- 4437.

Jacobs, S., C. Schilf, F. Fliegert, S. Kolling, Y. Weber, A. Schürmann, and H.-G. Joost. 1999. ADP-ribosylation factor (ARF)-like 4, 6, and 7 represent a subgroup of the ARF family characterized by rapid nucleotide exchange and a nuclear localization signal. *FEBS letters*. 456:384-388.

For supplement, below is the GTP exchange rate of Arl4 protein (data from Jacobs et al., 1999 FEBS letters):

Figure provided for reviewer has been removed. It showed Figure 6 from Jacobs, S., C. Schilf, F. Fliegert, S. Kolling, Y. Weber, A. Schürmann, and H.-G. Joost. 1999. ADPribosylation factor (ARF)-like 4, 6, and 7 represent a subgroup of the ARF family characterized by rapid nucleotide exchange and a nuclear localization signal. *FEBS letters*. 456:384-388. (doi: 10.1016/S0014-5793(99)00759-0)

6. Structural Compatibility of ARL4D and Rab GAPs Given the structural differences between ARFs and Rabs—particularly the N-terminal helical extension in ARFs—it is difficult to envision how ARL4D would interact with a Rab GAP domain. Structural modeling using AlphaFold to predict interactions between ARL4D, Rab7 (positive control), and the GAP domain of TBCD15 would provide valuable insight. Evidence supporting the structural feasibility of such a complex would significantly strengthen the manuscript. Future work could aim to obtain empirical structural data.

Response: We thank the reviewer for this insightful point regarding structural compatibility. To address it, we performed in silico modeling using AlphaFold 3 (Abramson et al., 2024) to evaluate potential interfaces between the TBC1D15 GAP domain (residues 326-562) and either Rab7 (positive control) or Arl4D. The models suggest possible contact regions for both Rab7-TBC1D15 and Arl4D-TBC1D15 pairs; however, the per-residue confidence metrics and overall model confidence scores were low (below ~0.5), indicating limited reliability of the predicted interaction interfaces. Additionally, AlphaFold Server terms restrict use of its outputs for automated ligand- or interaction-prediction workflows, which constrains our ability to rely on these models for definitive binding-site assignment. Therefore, while the preliminary modeling provides suggestive hypotheses, it does not furnish conclusive structural evidence that ARL4D can engage the Rab GAP domain of TBC1D15. We agree that obtaining empirical structural data (e.g., cryo-EM, X-ray crystallography, or NMR of the complex) would be the most rigorous way to resolve this question.

TBC1D15 326-562 a.a. sequence

Amino acids : alpha helices
 Amino acids : contact site (with Rab7)
 Amino acids : overlapping contact sites

```

1 IDSEGRILNVDNMKQMI FRGGLSHALRKQAWKFL LGYFPWDSTKEERTQL
51 QKQKTDEYFRMKLOWKS I SQEQEKRNRLRDYRSLIEKDVNRIDRTNKFY
101 EGQDN PGL I L L L HD I L M T Y C M Y D F D L G Y V Q G M S D L L S P L L Y V M E N E V D A F W
151 C F A S Y M D Q M H Q N F E E C M D G M K T Q L I Q L S T L L R L L D S G F C S Y L E S Q D S G Y L
201 Y F C F R W L L I R F K R E F S F L D I L R L W E V M W T E L P C T N F H
  
```

ipTM = 0.19
 pTM = 0.49

TBC1D15 326-562 a.a. sequence

Amino acids : alpha helices
 Amino acids : contact site (with Arl4D)
 Amino acids : overlapping contact sites

```

1 IDSEGRILNVDNMKQMI FRGGLSHALRKQAWKFL LGYFPWDSTKEERTQL
51 QKQKTDEYFRMKLOWKS I SQEQEKRNRLRDYRSLIEKDVNRIDRTNKFY
101 EGQDN PGL I L L L HD I L M T Y C M Y D F D L G Y V Q G M S D L L S P L L Y V M E N E V D A F W
151 C F A S Y M D Q M H Q N F E E C M D G M K T Q L I Q L S T L L R L L D S G F C S Y L E S Q D S G Y L
201 Y F C F R W L L I R F K R E F S F L D I L R L W E V M W T E L P C T N F H
  
```

ipTM = 0.21
 pTM = 0.47

Supplementary information on the analysis of the pTM and ipTM scores of AlphaFold 3 Molecular Modelling (<https://alphafoldserver.com/faq>) as follows:

The predicted Template Modeling (pTM) score and the Interface Predicted Template Modeling (ipTM) score are both derived from a measure called the Template Modeling (TM) score. This measures the accuracy of the entire structure (Zhang and Skolnick, 2004; Xu and Zhang, 2010). A pTM score above 0.5 means the overall predicted fold for the complex might be similar to the true structure. ipTM measures the accuracy of the predicted relative positions of the subunits within the complex. Values above 0.8 represent confident high-quality predictions, while values below 0.6 likely suggest a failed prediction. ipTM values between 0.6 and 0.8 are a gray zone where predictions could be correct or incorrect. The TM score is very stringent for small structures or short chains, so pTM assigns values below 0.05 when fewer than 20 tokens are involved; for these cases PAE or pLDDT may be more indicative of prediction quality.

References:

Abramson J, Adler J, Dunger J, Evans R, Green T, Pritzel A, Ronneberger O, Willmore L, Ballard AJ, Bambrick J, Bodenstein SW, Evans DA, Hung CC, O'Neill M, Reiman D, Tunyasuvunakool K, Wu Z, Žemgulytė A, Arvaniti E, Beattie C, Bertolli O, Bridgland A, Cherepanov A, Congreve M, Cowen-Rivers AI, Cowie A, Figurnov M, Fuchs FB, Gladman H, Jain R, Khan YA, Low CMR, Perlin K, Potapenko A, Savy P, Singh S, Stecula A, Thillaisundaram A, Tong C, Yakneen S, Zhong ED, Zielinski M, Židek A, Bapst V, Kohli P, Jaderberg M, Hassabis D, Jumper JM. Accurate structure prediction of biomolecular interactions with AlphaFold 3. *Nature*. 2024 Jun;630(8016):493-500. doi: 10.1038/s41586-024-07487-w.

Zhang Y, Skolnick J. Scoring function for automated assessment of protein structure template quality. *Proteins*. 2004 Dec 1;57(4):702-10. doi: 10.1002/prot.20264.

Xu J, Zhang Y. How significant is a protein structure similarity with TM-score = 0.5? *Bioinformatics*.

2010 Apr 1;26(7):889-95. doi: 10.1093/bioinformatics/btq066.

Minor comments

1. *Experimental Detail and Accessibility.* Although this is a short report, the experimental descriptions are minimal and may be challenging for non-specialists. For broader accessibility (e.g., JCS readership), the authors should expand on the experimental details in the main text. For example, in Figure 1a, the role of the mitochondrial marker Tim23 is not explained, making interpretation difficult.

Response: We thank the reviewer's comment for the feedback. To improve the problem of accessibility, we have modified our figure legend for Figure 1A as following: "Lysates from HeLa cells cultured for indicated duration were fractionated by gradient centrifugation and analyzed by immunoblotting. Fractions present with Tim23 and α -Tubulin indicates the mitochondria and cytosol fraction, respectively." We have also clarified the usage of each marker in the "Antibodies" section under Material and Methods.

2. *Terminology in Figure 3.* In the description of Figure 3b, the term "nullified" is too strong given the partial nature of the effect. "Partially restores" would be more appropriate. Similarly, in panel 3d, "partially restored" better reflects the data.

Response: The descriptions are modified as suggested: "Furthermore, co-incubation of Arl4D with TBC1D15 D397A, a GAP activity dead mutant (Rak et al., 2000; Yamano et al., 2014), partially restores GTP hydrolysis of Arl4D compared to TBC1D15 WT (Fig. 3B)." "We found depleting TBC1D15 under serum starvation partially restored Arl4D activity as the control groups. (Fig. 3D)."

Second decision letter

MS ID#: jcs.264304R1

MS TITLE: TBC1D15 functions as an Arl4D GAP and promotes mitochondrial translocation of Arl4D for organelle homeostasis

AUTHORS: Chia-Tang Chen; Tsai-Jung Liu; Shin-Jin Lin; Ting-Wei Chang; Fang-Jen S Lee

ARTICLE TYPE: Short Report

Dear Dr Lee,

We have now reached a decision on the above manuscript.

As you will see, the reviewers gave favourable reports but reviewer 1 raised some minor points that will require text and figure amendments to your manuscript. I hope that you will be able to carry these out because I would like to be able to accept your paper.

Reviewer 1

Advance summary and potential significance to field

I appreciate the revisions that the authors have made to the manuscript, which is much improved as a result. I have only minor criticisms for revision prior to publication.

1. The authors should show all replicates of their optiprep data (Fig. 1A, 1E, 2B, S3C), either in supplemental, or by quantifying all replicates and showing bar graphs in the figures.
2. The authors should include their data showing that Fis1-mediated recruitment of TBC1D15 does not recruit Arl4D to mitochondria in Supplementary Figure 3, as it supports two claims made in the manuscript: 1) that TBC1D15 is inactivating Arl4D at the plasma membrane, 2) that Arl4D-mediated mitophagy appears to be a distinct mechanism from that previously published regarding TBC1D15 recruitment to mitochondria.
3. Similarly, the authors should include their data that depletion of Rab7 does not trigger Arl4D translocation in the manuscript as well, as it is a key control showing that the effects of TBC1D15 on Arl4D translocation and mitophagy are not via inactivation of Rab7, supporting their claim that TBC1D15 is acting as a GAP to inactivate Arl4D, and that this is a new function for TBC1D15 independent of its well-established GAP activity against Rab7.
4. The authors should also mention in the manuscript that Arl4D translocates to mitochondria in CCCP-treated MDA-MB-231 cells, as repeating key findings in an independent cell line is an important control and strengthens the validity of their data, particularly given concerns regarding the physiological relevance of HeLa cells.
5. Page 2, line 50 states "whereas fragmentation indicates increased fission events," this is not true as fragmentation can also result from a decrease in the rate of mitochondrial fusion. Please revise.
6. On page 3, line 22, please include a brief explanation of the Pak-PBD pull-down assay, to increase readability of the manuscript for a broader mitochondria audience that is not necessarily familiar with Arl or Arf proteins.
7. Page 4, line 33, please change "The binding specificity was confirmed by in vivo Co-IP" to "The binding specificity was confirmed by Co-IP in cultured cells"
8. Page 4, lines 51 and 55, "colocalize with" should be "colocalizes with the"
9. Fig. 3D, please change the legend on the graph from "Relative pull-downed Arl4D" to "Relative Arl4D-GTP"

Reviewer 2

Advance summary and potential significance to field

The authors have adequately addressed my comments and the manuscript is improved.

Second revision

Author response to reviewers' comments

Reply to Reviewer #1

Reviewer #1:

I appreciate the revisions that the authors have made to the manuscript, which is much improved as a result. I have only minor criticisms for revision prior to publication.

1. The authors should show all replicates of their optiprep data (Fig. 1A, 1E, 2B, S3C), either in supplemental, or by quantifying all replicates and showing bar graphs in the figures.

Response: We thank the reviewer for this important point regarding data representation. As suggested, we have now quantified the data from three independent biological replicates for all fractionation experiments. These quantifications are now presented as bar graphs in the revised Figures 1A, 1E, 4B, and S3D. This process also led us to re-evaluate the representative image for Figure 1E. To ensure consistency with the quantitative data, we have replaced the original blot with a new, more representative image. We believe these changes significantly strengthen the manuscript.

Figure 1A

Figure 1E

Figure 4B

Figure S3D

2. The authors should include their data showing that Fis1-mediated recruitment of TBC1D15 does not recruit Arl4D to mitochondria in Supplementary Figure 3, as it supports two claims made in the manuscript: 1) that TBC1D15 is inactivating Arl4D at the plasma membrane, 2) that Arl4D-mediated mitophagy appears to be a distinct mechanism from that previously published regarding TBC1D15 recruitment to mitochondria.

Response: We thank the reviewer for this valuable suggestion to further support our claims. We agree that demonstrating the localization of Arl4D during Fis1-mediated TBC1D15 recruitment is critical for distinguishing this pathway from previously described mechanisms.

As requested, we have included new experimental data in **Supplementary Figure S3B**. While our previous revision included two independent experiments showing Fis1/TBC1D15 recruitment and Arl4D localization separately, we have replaced them with a single, more robust co-transfection experiment in the current manuscript. This was done to optimize space in Supplementary Figure S3 and to provide a more direct comparison by observing the effect of HA-Fis1 overexpression in cells co-transfected with myc-TBC1D15 and Arl4D.

These results clearly demonstrate that while Fis1 successfully recruits TBC1D15 to the mitochondria, it fails to co-recruit Arl4D. This finding reinforces our model that the Arl4D-TBC1D15 interaction is spatially distinct from mitochondrial Fis1-TBC1D15 signaling and likely occurs at the plasma membrane.

We have updated the main text to incorporate this finding:

“In contrast, recruitment of TBC1D15 to the mitochondria via Fis1 does not induce mitochondrial targeting of Arl4D, supporting a model where the Arl4D-TBC1D15 interaction occurs at a non-mitochondrial location (Fig. S3B).”

3. Similarly, the authors should include their data that depletion of Rab7 does not trigger Arl4D translocation in the manuscript as well, as it is a key control showing that the effects of TBC1D15 on Arl4D translocation and mitophagy are not via inactivation of Rab7, supporting their claim that TBC1D15 is acting as a GAP to inactivate Arl4D, and that this is a new function for TBC1D15 independent of its well-established GAP activity against Rab7.

Response: We thank the reviewer for this excellent suggestion. To directly test whether the effect of TBC1D15 on Arl4D is independent of its known GAP activity on Rab7, we performed the suggested control experiment. We have now included data showing that siRNA-mediated depletion of Rab7 does not trigger Arl4D translocation in the new Figure S4A. This result is crucial as it supports our claim that TBC1D15 acts on Arl4D, representing a new function independent of its established role with Rab7. We have revised the main text to incorporate this finding:

“Given that TBC1D15 is also the GAP for Rab7, an established mitophagy regulator, we tested whether Arl4D translocation was an indirect effect of Rab7 inactivation. However, siRNA-mediated knockdown of Rab7 had no effect on Arl4D localization, ruling out the involvement of Rab7 in this process (Fig S4A).”

4. The authors should also mention in the manuscript that Arl4D translocates to mitochondria in CCCP-treated MDA-MB-231 cells, as repeating key findings in an independent cell line is an important control and strengthens the validity of their data, particularly given concerns regarding the physiological relevance of HeLa cells.

Response: We thank the reviewer for this constructive suggestion. We agree that demonstrating Arl4D translocation in an independent cell line strengthens the physiological relevance and generalizability of our findings. As requested, we have included new experimental data using MDA-MB-231 cells treated with CCCP. This updated result, which includes improved representative

imaging and robust quantification, has been added as **Figure S1C**. We have updated the manuscript to reflect this addition. The revised text reads:

“..... promotes the translocation of Arl4D to the mitochondria in both HeLa and MDA-MB-231 cell lines, confirming that the response is not cell-type specific (Fig. S1B, C).”

5. Page 2, line 50 states “whereas fragmentation indicates increased fission events,” this is not true as fragmentation can also result from a decrease in the rate of mitochondrial fusion. Please revise.

Response: We are grateful to the reviewer for this crucial clarification. We have corrected this oversimplification in the manuscript. We now state that fragmentation could result from either accelerated fission or inhibited fusion and have applied this more precise definition to our findings. The revised text reads:

“The mitochondrial fragmentation we previously observed upon Arl4D-T35N localization (Li et al., 2012) indicates an imbalance in mitochondrial dynamics, which could result from either accelerated fission or inhibited fusion, while mitochondria from cells overexpressing Arl4D-WT and Arl4D-Q80L remain tubular (Fig. S2A).”

6. On page 3, line 22, please include a brief explanation of the Pak-PBD pull-down assay, to increase readability of the manuscript for a broader mitochondria audience that is not necessarily familiar with Arl or Arf proteins.

Response: We appreciate the reviewer’s suggestion to clarify this method. We have revised the sentence to integrate a brief explanation of the assay’s function:

“We then measured Arl4D activity using a Pak-PBD pull-down assay, which specifically isolates the active, GTP-bound form of Arl4 proteins (Chen et al., 2020). This experiment showed that serum starvation significantly decreased the amount of active Arl4D in HeLa cells.”

References:

Chen, K.-J., T.-C. Chiang, C.-J. Yu, and F.-J.S. Lee. 2020. Cooperative recruitment of Arl4A and Pak1 to the plasma membrane contributes to sustained Pak1 activation for cell migration. *Journal of cell science*. 133: jcs233361.

7. Page 4, line 33, please change “The binding specificity was confirmed by in vivo Co-IP” to “The binding specificity was confirmed by Co-IP in cultured cells”.

Response: We thank the reviewer for this important clarification. We have corrected the sentence as suggested. The revised text now reads: “The binding specificity was confirmed by Co-IP in cultured cells among Arl4 family and the interaction was also verified with endogenous levels of Arl4D.”

8. Page 4, lines 51 and 55, "colocalize with" should be "colocalizes with the".

Response: Thank you for catching these grammatical errors. Both sentences have been corrected in the revised manuscript as suggested.

"We next examined whether TBC1D15 colocalizes with the active form of Arl4D at the plasma membrane." "We found that TBC1D15 only colocalizes with the GTP-locked mutant of Arl4D at the plasma membrane (Fig. S3A), corresponding to the TBC1D15 binding preference of GTP-bound Arl4D."

9. Fig. 3D, please change the legend on the graph from "Relative pull-downed Arl4D" to "Relative Arl4D-GTP level".

Response: We agree that this change improves clarity. The legend on the graph in Figure 3D has been revised as requested.

Third decision letter

MS ID#: jcs.264304R2

MS Title: TBC1D15 functions as an Arl4D GAP and promotes mitochondrial translocation of Arl4D for organelle homeostasis

Authors: Chia-Tang Chen; Tsai-Jung Liu; Shin-Jin Lin; Ting-Wei Chang; Fang-Jen S Lee

Article Type: Short Report

Dear Dr Lee,

I am happy to tell you that your manuscript has been accepted for publication in Journal of Cell Science, pending standard publication integrity checks.